# A synaptotagmin suppressor screen indicates SNARE binding controls the timing and Ca$^{2+}$ cooperativity of vesicle fusion

Zhuo Guan[1,2,3], Maria Bykhovskaia[4], Ramon A Jorquera[5], Roger Bryan Sutton[6], Yulia Akbergenova[1,2,3], J Troy Littleton[1,2,3]*

[1]Picower Institute for Learning and Memory, Massachusetts Institute of Technology, Cambridge, United States; [2]Department of Biology, Massachusetts Institute of Technology, Cambridge, United States; [3]Department of Brain and Cognitive Sciences, Massachusetts Institute of Technology, Cambridge, United States; [4]Department of Neurology, School of Medicine, Wayne State University, Detroit, United States; [5]Neuroscience Department, Universidad Central del Caribe, Bayamon, Puerto Rico; [6]Department of Cell Physiology and Molecular Biophysics, Texas Tech University Health Sciences Center, Lubbock, United States

**Abstract** The synaptic vesicle Ca$^{2+}$ sensor Synaptotagmin binds Ca$^{2+}$ through its two C2 domains to trigger membrane interactions. Beyond membrane insertion by the C2 domains, other requirements for Synaptotagmin activity are still being elucidated. To identify key residues within Synaptotagmin required for vesicle cycling, we took advantage of observations that mutations in the C2B domain Ca$^{2+}$-binding pocket dominantly disrupt release from invertebrates to humans. We performed an intragenic screen for suppressors of lethality induced by expression of Synaptotagmin C2B Ca$^{2+}$-binding mutants in *Drosophila*. This screen uncovered essential residues within Synaptotagmin that suggest a structural basis for several activities required for fusion, including a C2B surface implicated in SNARE complex interaction that is required for rapid synchronization and Ca$^{2+}$ cooperativity of vesicle release. Using electrophysiological, morphological and computational characterization of these mutants, we propose a sequence of molecular interactions mediated by Synaptotagmin that promote Ca$^{2+}$ activation of the synaptic vesicle fusion machinery.

DOI: https://doi.org/10.7554/eLife.28409.001

*For correspondence: troy@mit.edu

**Competing interests:** The authors declare that no competing interests exist.

## Introduction

Neurotransmitter release requires the rapid fusion of synaptic vesicles in response to Ca$^{2+}$ influx following an action potential. The SNARE complex is a highly conserved protein machinery that controls the fusion of synaptic vesicles with the plasma membrane (*Südhof and Rothman, 2009*). Beyond the SNARE complex, neurons have evolved additional mechanisms to regulate the timing and localization of fusion. Among these key regulators is the Synaptotagmin (Syt) family of Ca$^{2+}$ sensors (*Chapman, 2008*; *Südhof, 2012*; *Yoshihara et al., 2003*). Synaptotagmin 1 (Syt1) functions as the Ca$^{2+}$ sensor for driving synchronous fusion of synaptic vesicles following an action potential (*Geppert et al., 1994*; *Yoshihara and Littleton, 2002*), with other isoforms like Syt7 regulating distinct steps in vesicle cycling (*Bacaj et al., 2013*; *Jackman et al., 2016*; *Liu et al., 2014*; *Wen et al., 2010*). Syt1 is tethered to synaptic vesicles through a single transmembrane domain and contains two cytosolic Ca$^{2+}$ binding C2 domains (*Perin et al., 1990*). The C2 domains, termed C2A and C2B,

contain $Ca^{2+}$-binding loops that emerge from the top of each domain and insert into negatively charged lipid bilayers following $Ca^{2+}$ influx (*Chapman and Davis, 1998*; *Chapman and Jahn, 1994*; *Davletov and Südhof, 1993*; *Desai et al., 2000*; *Fernandez et al., 2001*). Beyond lipid interactions, the mechanistic details of how Syt1 regulates fusion are still being elucidated. Syt1 can interact with the SNARE complex, with several studies suggesting an important role for this interaction during fusion (*Bennett et al., 1992*; *Brewer et al., 2015*; *Chicka et al., 2008*; *Choi et al., 2010*; *Li et al., 1995*; *Littleton et al., 2001*; *Lynch et al., 2008*; *Lyubimov et al., 2016*; *Pang et al., 2006*; *Schupp et al., 2016*; *Zhou et al., 2015*). However, not all studies support a Syt1-SNARE binding mechanism during release (*Jahn and Fasshauer, 2012*; *Park et al., 2015*). Syt1 also contains a $Ca^{2+}$-independent lipid-binding surface on one side of the C2B domain that represents a candidate mechanism for docking Syt1 on the plasma membrane, potentially positioning the $Ca^{2+}$ binding loops near sites where fusion will occur (*Bai et al., 2004*; *Kuo et al., 2009*; *Loewen et al., 2006*; *Mackler and Reist, 2001*; *Pérez-Lara et al., 2016*; *Rickman et al., 2004*; *Wang et al., 2016*; *Young and Neher, 2009*). In addition, Syt1 undergoes multimerization (*Chapman et al., 1998*; *Fukuda and Mikoshiba, 2000*; *Lee et al., 2013*; *Littleton et al., 2001*; *Wu et al., 2003*; *Zanetti et al., 2016*), though it is unknown whether a Syt1 monomer or oligomer represents the active state of the protein. Syt1 is abundant on synaptic vesicles, with estimates of the copy number per vesicle ranging from 7 to 15 (*Mutch et al., 2011*; *Takamori et al., 2006*). It is possible that Syt1 cycles between distinct conformations and multimeric states, but such mechanisms are unknown. A large-scale identification of key residues within Syt1 that coordinate its activity would represent an important step towards characterizing how Syt1 operates during the fusion cycle.

Prior work identified $Ca^{2+}$ binding to the Syt1 C2B domain as a critical step required for fast synchronous release, with $Ca^{2+}$ binding to C2A playing an important but more modulatory role in the process (*Fernández-Chacón et al., 2001*; *Lee et al., 2013*; *Mackler et al., 2002*; *Nishiki and Augustine, 2004*; *Shin et al., 2009*; *Yoshihara et al., 2010*). The C2B $Ca^{2+}$ binding pocket emerges from the top of the domain and contains five negatively charged aspartate residues (numbered D1 to D5) that help coordinate binding of multiple $Ca^{2+}$ ions (*Fernandez et al., 2001*). *Drosophila* strains with mutations in one or more of the D1 to D5 residues in the C2B $Ca^{2+}$ binding pocket result in a dominant-negative (DN) phenotype, disrupting neurotransmission even in the presence of wild-type Syt1 (*Lee et al., 2013*; *Mackler et al., 2002*; *Yoshihara et al., 2010*). More recently, we identified autosomal dominant mutations in the $Ca^{2+}$ binding pocket of the C2B domain of human Syt2, a Syt1 homolog enriched in the PNS, that result in an autosomal-dominant disorder presenting with peripheral neuropathy and dysfunctional synaptic transmission at neuromuscular junctions (NMJs) (*Herrmann et al., 2014*; *Whittaker et al., 2015*). An autosomal-dominant mutation in the Syt1 C2B domain has also been found in a patient presenting with a more severe neurodevelopmental disorder (*Baker et al., 2015*). As such, mutations in the $Ca^{2+}$ binding pocket of C2B (hereafter referred to as C2B DN) result in a dominant disruption of synaptic transmission from invertebrates to humans, potentially through oligomerization with native Syt1 and poisoning of the fusion machinery.

The pupal lethality observed following overexpression of UAS-Syt1 C2B DN transgenes in *Drosophila* allowed us to perform a large-scale F1 ethyl methane sulfonate (EMS) screen for intragenic suppressors. A second site mutation in the Syt1 DN transgene that disrupted the ability of the protein to function properly should abolish or reduce the DN phenotype and allow animals expressing the transgene to survive. This screening approach allowed us to generate a large collection of new mutations in Syt1, with many lines containing mutations of the same amino acid multiple times in independent suppressors. Twenty-two essential residues within Syt1 were identified that are required for the dominant-negative activities of the mutant C2B protein. We characterized mutations that did not disrupt the stability of Syt1 and that mapped to distinct regions that suggested a structural basis for several activities required for vesicle release. These mutations were introduced into the native Syt1 protein in the absence of any C2B $Ca^{2+}$ binding mutation, allowing us to test their role in normal Syt1 function through electrophysiological, electron microscopy and computational approaches. Our data indicate intra-domain C2A-C2B interactions, the C2B polybasic region, and Synaptotagmin multimerization are important for promoting normal amounts of vesicle fusion, but are not critical for the timing of exocytosis. In contrast, we find a distinct C2B surface opposite the polybasic stretch that is implicated in SNARE complex interactions and required for rapid synchronization and $Ca^{2+}$ cooperativity of synaptic vesicle fusion.

## Results

### An intragenic screen for suppressors of Syt1 C2B dominant-negative mutations

Syt1 contains two C2 domains composed of conserved $Ca^{2+}$-binding pockets with five negatively charged aspartic acid (or glutamic acid) residues (termed D1 to D5) providing critical sites for $Ca^{2+}$-mediated lipid interactions (*Figure 1A*). The neutralization of subsets of these negatively charged $Ca^{2+}$-binding residues from aspartate to asparagine in the C2B domain results in a DN disruption of synaptic transmission in *Drosophila* (*Lee et al., 2013*; *Mackler et al., 2002*; *Yoshihara et al., 2010*). Pan-neuronal overexpression of UAS-Syt1C2B[D356N,D362N] (referred to hereafter as Syt1C2B[D1,2N]) or UAS-Syt1C2B[D416N,D418N] (referred to hereafter as Syt1C2B[D3,4N]) transgenes with elav[C155] GAL4 in otherwise wildtype *Drosophila* with two endogenous copies of native Syt1 causes late pupal lethality. Animals expressing the transgene undergo normal metamorphosis, but are too weak to emerge from the pupal case. Manual removal of adults from the pupal case revealed severe defects in locomotion, with animals unable to coordinate normal walking or flight. To characterize the underlying defects in synaptic transmission, we performed two-electrode voltage clamp recordings from third instar larvae overexpressing the DN transgenes with elav[C155] (*Figure 1B–E*). We focused on strains overexpressing Syt1C2B[D1,2N], but observed similar results following overexpression of Syt1C2B[D3,4N] (*Lee et al., 2013*) or Syt1C2B[D2A] (*Herrmann et al., 2014*). Overexpression of DN Syt1 resulted in a strong suppression of the evoked response, reducing the evoked excitatory junctional current (eEJC) by 86% compared to controls (*white*) (*Figure 1B,C*). In contrast, overexpression of wildtype Syt1 lacking these mutations did not reduce eEJCs. In addition to the strong suppression of synchronous evoked release, the C2B DN mutations also enhanced the rate of spontaneous fusion, increasing mini frequency more than two-fold (*Figure 1D,E*). The strong reduction in evoked release and elevated mini frequency induced by Syt1C2B[D1,2N] overexpression in the presence of two endogenous copies of the Syt1 locus largely mimics the effects on synaptic transmission observed in *Drosophila* *syt1* null mutants (*Jorquera et al., 2012*; *Lee et al., 2013*; *Littleton et al., 1994*; *1993*; *Yoshihara and Littleton, 2002*), suggesting the DN transgenic protein may multimerize with and inactivate endogenous Syt1. Alternatively, the DN protein may poison the fusion machinery through other abnormal interactions that disable the release apparatus in a way similar to the loss of native Syt1.

To gain a better understanding of how C2B $Ca^{2+}$ binding mutations disrupt neurotransmitter release, we performed an intragenic EMS suppressor screen to identify key residues in an unbiased fashion (*Figure 1F*). Males harboring the toxic Syt1 transgene (Syt1 C2B[D1,2N] or Syt1 C2B[D3,4N]) on the third chromosome were mutagenized with EMS and crossed to elav[C155] GAL4 females. All F1 progeny from this cross are normally pharate adult lethal. Any mutation that suppressed the ability of the Syt1 C2B $Ca^{2+}$ binding mutant to exert a DN effect on synaptic function would be expected to generate viable F1 progeny. This provided a fast and robust F1 screening platform to identify important Syt1 residues on a large-scale. The screen could also recover dominant extragenic suppressors that targeted other loci, but we focused only on intragenic suppressors for the current study. Any viable progeny were mated to a third chromosome balancer line to capture the mutated chromosome. Genomic DNA was obtained from progeny of the viable F1 generation and the Syt1 transgene was PCR amplified and sequenced. We screened over 500,000 F1 offspring and identified 105 mutations that produced viable progeny (*Table 1*). Sequence analysis confirmed 54 of the viable lines contained a premature stop codon that was induced by EMS within the Syt1 transgene, providing an explanation for how the DN effects were eliminated in this subset of suppressors that fail to express the full-length protein. In addition to these stop codons, we identified an additional 22 point mutations present throughout the C2A and C2B domains that resulted in amino acid substitutions that led to viable progeny (*Figure 1G–J*). We subdivided these point mutations into those that destabilized the protein, leading to reduced expression, versus those that did not alter Syt1 levels by Western analysis (*Figure 1—figure supplement 1*). Over half of the point mutations identified reduced expression when assayed by Western blotting, suggesting they alter protein stability (*Figure 1—figure supplement 1*). To facilitate mapping of the mutations, we created a homology model of *Drosophila* Syt1 (*Figure 1H–J*) based on the strong conservation with the previously elucidated structure of human Syt1 (*Fuson et al., 2007*). Throughout the remaining figures, the C2A domain is indicated in blue, the C2B domain in magenta, and the $Ca^{2+}$ binding loops in green. Many of the

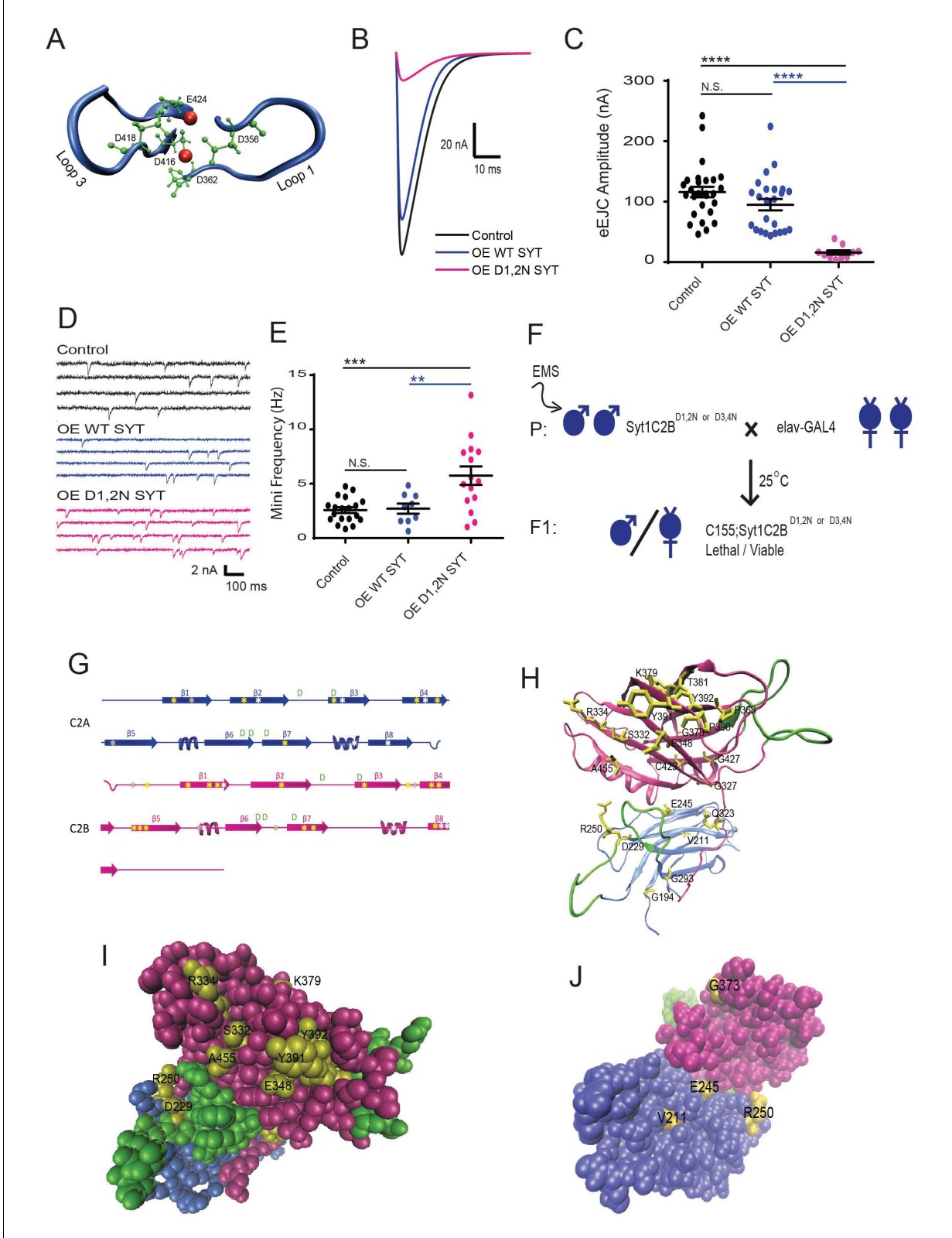

**Figure 1.** An intragenic suppressor screen for mutations that disrupt the lethality induced by Syt1C2B[D1,2N] or Syt1C2B[D3,4N] expression. (**A**) View of key residues in the Syt1 C2B $Ca^{2+}$ binding pocket. The $Ca^{2+}$ binding loops 1 and 3 are highlighted in blue, the negatively charged $Ca^{2+}$-binding residues in green, and $Ca^{2+}$ ions in red. Neuronal overexpression of Syt1C2B[D416,D418N (D1,2N)] or Syt1C2B[D356,D362N (D3,4N)] results in pharate adult lethality. (**B**) Representative evoked excitatory junctional currents (eEJCs) recorded in 0.2 mM extracellular $Ca^{2+}$ in control *white* larvae (black trace), elav[C155]-GAL4;

*Figure 1 continued on next page*

Figure 1 continued

UAS-Syt1 wildtype (OE WT SYT, blue trace) and elav$^{C155}$-GAL4; UAS-Syt1C2B$^{D1,2N}$ (OE D1,2N SYT, magenta trace). (C) Quantification of mean eEJC amplitude in the indicated genotypes: control, $116.0 \pm 8.7$ nA, n = 27; OE WT Syt1, $95.0 \pm 9.4$ nA, n = 24; OE D1,2N Syt1, $15.9 \pm 3.5$ nA, n = 10. (D) Representative postsynaptic current recordings of spontaneous release in the indicated genotypes. (E) Quantification of average mini frequency for the indicated genotypes: control, $2.6 \pm 0.3$ Hz, n = 19; OE WT Syt1, $2.7 \pm 0.5$ Hz, n = 9; OE D1, 2N Syt1, $5.7 \pm 0.8$ Hz, n = 15. (F) Crossing scheme used for EMS screening of suppressors of Syt1C2B$^{D1,2N \text{ or } D3,4N}$-induced lethality. (G) Location of identified suppressors (point mutant - yellow asterisks, stop codon - white asterisks, deletion - grey asterisks) on the Syt1 secondary structure. C2A is indicated blue and C2B in magenta. (H) View of identified point mutant alleles (yellow) in the DSyt1 homology model. For all remaining structural images, the C2A domain is colored blue and the C2B domain is colored magenta, while the Ca$^{2+}$ binding loops are highlighted in green. (I) Suppressor point mutants (yellow) located in a space-filling model of DSyt1. (J) Suppressor mutations (yellow) located on the opposite face of the DSyt1 structure compared to panel I. Statistical significance was determined using one-way ANOVA (nonparametric) with post hoc Sidak's multiple comparisons test. N.S. = no significant change, *p<0.05, **p<0.01, ***p≤0.0005, ****p<0.0001. All error bars are standard error of the mean (SEM).

DOI: https://doi.org/10.7554/eLife.28409.002

The following source data and figure supplement are available for figure 1:

**Source data 1.** Sample size (n), mean, SEM, and One-way Anova (and nonparametric) Sidak's multiple comparisons test are presented for the data in *Figure 1C* and *Figure 2C,G,K,O*.

DOI: https://doi.org/10.7554/eLife.28409.004

**Source data 2.** Sample size (n), mean, SEM, and One-way Anova (and nonparametric) Sidak's multiple comparisons test are presented for the data in *Figure 1E* and *Figure 2D,H,L and P*.

DOI: https://doi.org/10.7554/eLife.28409.005

**Source data 3.** Sample size (n), mean, SEM, and Student's t test are presented for the data in *Figure 1—figure supplement 1B, C and E*

DOI: https://doi.org/10.7554/eLife.28409.006

**Figure supplement 1.** Summary of Western analysis of Syt1 alleles that suppress lethality following overexpression of Syt1C2B$^{D1,2N}$ or Syt1C2B$^{D3,4N}$.

DOI: https://doi.org/10.7554/eLife.28409.003

mutated residues that reduce DN Syt1 protein levels are embedded in the interior of the protein (i.e. V211 and E245 in *Figure 1J*), indicating they likely disrupt folding or stability of Syt1. Ten of the remaining amino acid substitutions identified in multiple suppressor lines did not alter the levels of DN Syt1 protein (*Figure 1—figure supplement 1*). We focused on this category and hypothesized that mutation of these individual residues disrupt key Syt1 interactions required for the DN effects of the C2B Ca$^{2+}$ binding mutant.

## Intragenic suppressors in Syt1 C2B$^{D1,2N}$ rescue evoked release defects

To determine if the ability of intragenic suppressors to rescue viability is due to protective effects on neurotransmitter release, we analyzed if they altered synaptic physiology compared to overexpression of the UAS-Syt1 C2B$^{D1,2N}$ transgene alone (*Figure 2*). We focused on specific sets of residues that reside in distinct regions of Syt1 (*Figure 2A,E,I,M*). We identified several residue changes within the C2A domain that rescued viability (*Table 1*), including two hits to a previously characterized residue in the C2A Ca$^{2+}$ binding pocket (D229N) that disrupts C2A Ca$^{2+}$ binding (*Robinson et al., 2002*). We previously found that overexpression of Syt1 with mutations in both the C2A and C2B Ca$^{2+}$ binding sites did not cause dominant-negative defects in release (*Lee et al., 2013*). Together with the C2A D229N hits from this screen, these data indicate that intact Ca$^{2+}$ binding by C2A is required for the DN effects of the C2B domain mutations within *Drosophila*. This contrasts with observations for DN mammalian Syt1, where dual mutations in C2A and C2B do disrupt release (*Wu et al., 2017*). Although the basis of this species-specific difference is unclear, the C2A domain from *Drosophila* may be more essential for proper positioning of the DN C2B domain to poison the fusion machinery, a property disrupted by simultaneous loss of C2A Ca$^{2+}$-activated lipid binding.

Beyond the C2A Ca$^{2+}$ binding pocket, three C2A suppressor mutations altered residue R250 (R250H, R250C) located at the interface between the C2A and C2B domains, and near the site of a potential dimer interface from the crystal structure (*Figure 2A*). To determine if R250H disrupted the effects of the DN C2B Ca$^{2+}$ binding mutation on synaptic transmission, we performed physiological recording following Syt1 C2A$^{R250H}$ C2B$^{D1,2N}$ overexpression. The R250H mutation completely suppressed the DN effects, with a full recovery of the evoked response to control levels (*Figure 2B–C*). In contrast, the R250H mutation did not significantly rescue the enhanced spontaneous release (*Figure 2D*, *Figure 2—figure supplement 1*).

**Table 1.** Summary of identified alleles that suppress lethality following overexpression of Syt1C2B$^{D1,2N}$ or Syt1C2B$^{D3,4N}$.

The location of the residues is indicated for *Drosophila melanogaster*, and for the homologous rodent (*Rattus norvegicus*) residues. The number of independent suppressor hits for each residue is indicated, as well as the location on the Syt1 secondary structure. The result of Western analysis to determine stability of the overexpressed mutant protein is also indicated (D (degraded), S (stable) and NT (not tested)).

| Dros. AA# Change | (Rat AA#) | # Hits | Location | Stability |
|---|---|---|---|---|
| Q48 stop codon | | 1 | Intravesicular | NT |
| Q50 stop codon | | 5 | Intravesicular | NT |
| Q63 stop codon | | 1 | Intravesicular | NT |
| Q67 stop codon | | 3 | Intravesicular | NT |
| Q80 stop codon | | 8 | Intravesicular | NT |
| W113 stop codon | | 5 | Transmembrane | NT |
| R141 stop codon | | 2 | Linker domain | NT |
| K147 stop codon | | 1 | Linker domain | NT |
| Q158R/stop codon | | 2 | Linker domain | NT |
| Q191 stop codon | | 1 | Linker domain | NT |
| G194E | (G143) | 1 | C2A beta 1 | NT |
| N205 deletion | | 2 | C2A beta 1 | NT |
| V211E | (V160) | 1 | C2A beta 2 | D |
| Q215 stop codon | (Q164) | 1 | C2A beta 2 | NT |
| D229N | (D178) | 2 | C2A beta 3 | NT |
| Y231 stop codon | (Y180) | 1 | C2A beta 3 | NT |
| E245K | (E194) | 1 | C2A beta 4 | D |
| K247 stop codon | (K196) | 1 | C2A beta 4 | NT |
| R250H/C | (R199) | 3 | C2A beta 4 | S |
| P267 deletion | (P215) | 1 | C2A beta 5 | NT |
| G293D | (G241) | 1 | C2A beta 7 | NT |
| Q306 stop codon | | 3 | C2A loop between beta 7 & 8 | NT |
| W111 stop codon | (W259) | 3 | C2A beta 8 | NT |
| G321 deletion | | 1 | Linker between C2A & C2B | NT |
| Q323K/stop codon | (Q270) | 3 | Linker between C2A & C2B | NT |
| G327R | (G274) | 1 | C2B beta 1 | NT |
| S332L | (S279) | 2 | C2B beta 1 | S |
| R334H/C/L | (R281) | 6 | C2B beta 1 | S |
| Y335 stop codon | (Y282) | 1 | C2B beta 1 | NT |
| E348K | (E295) | 4 | C2B beta 2 | S |
| P363S | (P310) | 2 | C2B beta 3 | S |
| G373D | (G320) | 1 | C2B loop between beta 3 & 4 | S |
| R375 deletion | (R322) | 1 | C2B loop between beta 3 & 4 | NT |
| K379E | (K326) | 1 | C2B beta 4 | S |
| T381I/stop codon | (T328) | 3 | C2B beta 4 | D |
| P390L | (P337) | 2 | C2B beta 5 | D |
| Y391D/N/H | (Y338) | 4 | C2B beta 5 | S |
| Y392N | (Y339) | 1 | C2B beta 5 | S |
| Q404 deletion | (Q351) | 1 | C2B loop between beta 5 & 6 | NT |

*Table 1 continued on next page*

Table 1 continued

| Dros. AA# Change | (Rat AA#) | # Hits | Location | Stability |
|---|---|---|---|---|
| I405 stop codon(I352) | | 1 | C2B loop between beta 5 & 6 | NT |
| G421 deletion | (G368) | 1 | C2B loop between beta 6 & 7 | NT |
| G427D | (G374) | 5 | C2B beta 7 | Partial D |
| C429S | (V376) | 1 | C2B beta 7 | NT |
| W443 stop codon | (W390) | 2 | C2B loop between beta 7 & 8 | NT |
| A455T | (A402) | 3 | C2B beta 8 | S |
| Q456 stop codon | (Q403) | 2 | C2B beta 8 | NT |
| W457 stop codon | (W404) | 6 | C2B beta 8 | NT |

DOI: https://doi.org/10.7554/eLife.28409.007

A second group of four suppressors were identified that map onto the polybasic surface of the C2B domain, including K379E and T381 (*Figure 2E*). The K379 residue in particular has been suggested to function in $Ca^{2+}$ independent lipid binding that may pre-position Syt1 on the plasma membrane prior to fusion (*Loewen et al., 2006*; *Rickman et al., 2004*), and as part of a secondary SNARE binding surface (*Brewer et al., 2015*). The K379E mutation resulted in a partial suppression of the evoked response defect (*Figure 2F–G*), but had no significant effect on the enhanced spontaneous release (*Figure 2H*). These results suggest that docking of the Syt1 C2B domain onto the membrane prior to $Ca^{2+}$ influx is important for the Syt1C2B$^{D1,2N}$ mutation to fully disrupt fusion.

A third category of suppressors was defined by two mutations in P363 (P363S), which maps next to the C2B D2 (D362) $Ca^{2+}$ binding reside (*Figure 2I*). A mutation in the homologous residue (P308L) in human Syt2 has been found to cause an autosomal-dominant form of Lambert-Eaton Myasthenic Syndrome in a multigenerational UK family (*Herrmann et al., 2014*). The P363S mutation partially suppressed the defective evoked response following expression of C2B$^{D1,2N}$ (*Figure 2J–K*) and had no significant effect on the enhanced spontaneous release (Fig, 2L). Given P363 resides near the C2B $Ca^{2+}$ binding pocket, we hypothesize this mutation may alter the interaction of the C2B $Ca^{2+}$ binding loops with membranes or $Ca^{2+}$ to prevent the DN effects on release.

A final group of suppressor mutations identified in the screen mapped to a poorly characterized side of the C2B surface on the opposite face from the polybasic region (*Figure 2M*). We identified 19 suppressor mutations that disrupted five amino acids (S332, R334, E348, Y391 and A455) that decorate this C2B domain surface. A recent structure of the SNARE complex bound to Syt1 (*Zhou et al., 2015*) indicates these five residues contribute to the primary interface surface formed between the C2B domain and the SNARE complex (see below). We characterized two of these mutations in more detail (S332L and R334H). The R334H change completely suppressed the ability of the C2B$^{D1,2N}$ mutant to disrupt evoked release, while S332L partially suppressed the evoked defect (*Figure 2M–O*). Neither mutation was capable of significantly suppressing the increased spontaneous release observed following expression of C2B$^{D1,2N}$ (*Figure 2P*). These findings indicate that R334 plays an essential function in allowing the DN effects of the C2B $Ca^{2+}$ binding mutation to be manifested, with S332 contributing slightly less to the process.

In contrast to the ability of Syt1C2B$^{D1,2N}$ to dominantly disrupt release, overexpression of transgenes harboring each of the mutations described above (R250H, K379E, P363S, S332L and R334H) in the absence of the C2B $Ca^{2+}$ binding mutation failed to cause a dominant disruption of synaptic transmission on their own (*Figure 2—figure supplement 2*). In summary, each of the intragenic suppressors rescue viability by blocking the DN effects the C2B $Ca^{2+}$ binding mutation on evoked synaptic transmission, but do not significantly alter the effects on spontaneous fusion or disrupt release on their own when overexpressed.

## The R250H mutation disrupts evoked release and clamping of spontaneous fusion, but does not regulate release timing

In the absence of Syt1, multiple defects in neurotransmitter release occur at *Drosophila* NMJs. Synchronous synaptic vesicle release is dramatically reduced, asynchronous release is increased, and

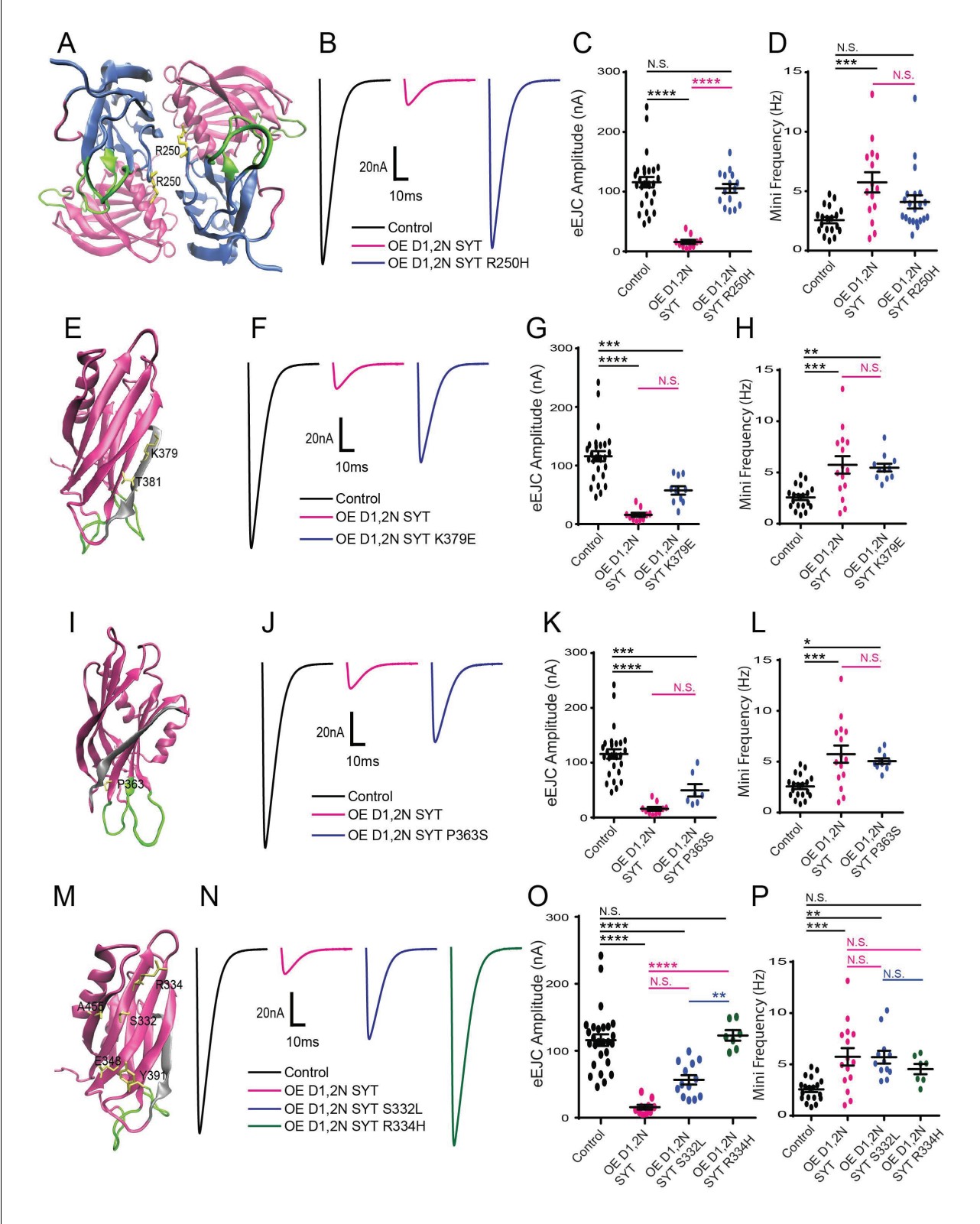

**Figure 2.** Characterization of suppressor mutation effects on DN Syt1C2B[D1,2N] physiology. (**A**) Location of the R250 residue (yellow) in C2A at the Syt1 dimer interface. (**B**) Representative eEJCs recorded in 0.2 mM extracellular Ca[2+] in control (black), elav[C155]-GAL4; UAS-Syt1C2B[D1,2N] (OE D1,2N SYT, magenta) and elav[C155]-GAL4; UAS-Syt1C2B[D1,2N R250H] (OE D1,2 N SYT R250H, blue). (**C**) Quantification of mean eEJC amplitude for the indicated genotypes: control, 116.0 ± 8.7 nA, n = 27; OE D1,2N Syt1, 15.9 ± 3.5 nA, n = 10; OE D1,2 N Syt1 R250H, 105.5 ± 7.5 nA, n = 15. (**D**) Quantification of

*Figure 2 continued on next page*

*Figure 2 continued*

average mini frequency for the indicated genotypes: control, 2.6 ± 0.3 Hz, n = 19; OE D1,2N Syt1, 5.7 ± 0.8 Hz, n = 15; OE D1,2N Syt1 R250H, 4.1 ± 0.6 Hz, n = 21. (E) Location of the K379 and T381 residues (yellow) in the C2B polybasic β-strand (grey). (F) Representative eEJCs recorded in 0.2 mM extracellular $Ca^{2+}$ in control (black), elav$^{C155}$-GAL4; UAS-Syt1C2B$^{D1,2N}$ (OE D1,2N SYT in magenta) and elav$^{C155}$-GAL4; UAS-Syt1C2B$^{D1,2N\ K379E}$ (OE D1,2N SYT K379E, blue). (G) Quantification of mean eEJC amplitude in the indicated genotypes: control, 116.0 ± 8.7 nA, n = 27; OE D1,2N Syt1, 15.9 ± 3.5 nA, n = 10; OE D1,2N Syt1 K379E, 57.7 ± 7.3 nA, n = 10. (H) Quantification of average mini frequency for the indicated genotypes: control, 2.6 ± 0.3 Hz, n = 19; OE D1,2N Syt1, 5.7 ± 0.8 Hz, n = 15; OE D1,2N Syt1 K379E, 5.5 ± 0.4 Hz, n = 11. (I) Location of the P363 residue (yellow) near the C2B $Ca^{2+}$ binding loops (green). (J) Representative eEJCs recorded in 0.2 mM extracellular $Ca^{2+}$ in control (black), elav$^{C155}$-GAL4; UAS-Syt1C2B$^{D1,2N}$ (OE D1,2N SYT, magenta) and elav$^{C155}$-GAL4; UAS-Syt1C2B$^{D1,2N\ K379E}$ (OE D1,2N SYT P363S, blue). (K) Quantification of mean eEJC amplitude in the indicated genotypes: control, 116.0 ± 8.7 nA, n = 27; OE D1,2N Syt1, 15.9 ± 3.5 nA, n = 10; OE D1,2N Syt1 P363S, 49.6 ± 11.2 nA, n = 7. (L) Quantification of average mini frequency for the indicated genotypes: control, 2.6 ± 0.3 Hz, n = 19; OE D1,2N Syt1, 5.7 ± 0.8 Hz, n = 15; OE D1,2N Syt1 P363S, 5.1 ± 0.3 Hz, n = 10. (M) Location of the S332, R334, E348, Y391, and A455 residues (yellow) on the C2B surface opposite to the polybasic β-strand (grey). (N) Representative eEJCs recorded in 0.2 mM extracellular $Ca^{2+}$ in control (black), elav$^{C155}$-GAL4; UAS-Syt1C2B$^{D1,2N}$ (OE D1,2N SYT, magenta), elav$^{C155}$-GAL4; UAS-Syt1C2B$^{D1,2N\ S332L}$ (OE D1,2N SYT S332L, blue) and elav$^{C155}$-GAL4; UAS-Syt1C2B$^{D1,2N\ R334H}$ (OE D1,2N SYT R334H, green). (O) Quantification of mean eEJC amplitude in the indicated genotypes: control, 116.0 ± 8.7 nA, n = 27; OE D1,2N Syt1, 15.9 ± 3.5 nA, n = 10; OE D1,2N Syt1 S332L, 56.6 ± 7.0 nA, n = 13; OE D1,2N Syt1 R334H, 122.9 ± 7.8 nA, n = 7. (P) Quantification of average mini frequency for the indicated genotypes: control, 2.6 ± 0.3 Hz, n = 19; OE D1,2N Syt1, 5.7 ± 0.8 Hz, n = 15; OE D1,2N Syt1 S332L, 5.7 ± 0.6 Hz, n = 12; OE D1,2N Syt1 R334H, 4.5 ± 0.5 Hz, n = 7. Statistical significance was determined using one-way ANOVA (nonparametric) with post hoc Sidak's multiple comparisons test. N.S. = no significant change, *p<0.05, **=p<0.005, ***=p<0.0005, ****=p<0.0001. All error bars are SEM.

DOI: https://doi.org/10.7554/eLife.28409.008

The following source data and figure supplements are available for figure 2:

**Source data 1.** Sample size (n), mean, SEM, and One-way Anova (and nonparametric) Turkey's multiple comparisons test are presented for the data in *Figure 2—figure supplement 2C,D,F,G,I,J,L,M*

DOI: https://doi.org/10.7554/eLife.28409.011

**Figure supplement 1.** Postsynaptic current recordings of spontaneous release at third instar larval muscle 6 synapses in controls and in larvae overexpessing the indicated Syt1 transgenic protein by elav-GAL4 in HL3.1 saline.

DOI: https://doi.org/10.7554/eLife.28409.009

**Figure supplement 2.** Characterization of the effects of mutant overexpression in the wildtype background.

DOI: https://doi.org/10.7554/eLife.28409.010

spontaneous fusion rates rise (*Broadie et al., 1994*; *DiAntonio and Schwarz, 1994*; *Littleton et al., 1994*; *1993*; *Yoshihara and Littleton, 2002*). In addition, there are defects in synaptic vesicle docking (*Reist et al., 1998*), endocytosis (*Littleton et al., 2001*; *Poskanzer et al., 2006*; *2003*) and $Ca^{2+}$ cooperativity of fusion (*Littleton et al., 1994*; *Yoshihara and Littleton, 2002*). These results indicate Syt1 plays a multi-functional role in the synaptic vesicle cycle and regulates docking and endocytosis, activates fast synchronous release as the $Ca^{2+}$ sensor, suppresses slower asynchronous release and acts to prevent spontaneous vesicle fusion. We reasoned that the collection of point mutants from the screen would provide a powerful genetic toolbox to begin to decipher which residues (and by extension, potential binding partners) regulate each of these steps of the synaptic vesicle cycle.

To determine how the point mutations alter the normal activity of Syt1 independent of the DN mutations in the C2B $Ca^{2+}$ binding pocket, we cloned the mutations into the wild type Syt1 transgene. We generated rescue lines expressing the individual point mutants in the *syt1* null background. To ensure equal expression among the transgenes and eliminate any genomic position effects, we employed site-specific transformation via the ΦC31 integrase system for transgene insertion (*Groth et al., 2004*). Immunostaining for Syt1 in third instar larvae of the transgenic rescues indicated each mutant protein was expressed normally at synaptic terminals compared to controls (*Figure 3—figure supplement 1A–D*), indicating no defect in synaptic localization at this level of resolution. The transgenic mutant proteins were expressed at similar levels to the wildtype Syt1 transgenic rescue by western analysis (*Figure 3—figure supplement 1E*), though lower that endogenous Syt1 (*Figure 3—figure supplement 2A*). The reduced levels of Syt1 in the wildtype transgenic rescue resulted in less overall evoked vesicle fusion (*Figure 3—figure supplement 2B,C*), though release was fully synchronous as seen in controls. As such, we compared all the transgenic rescue lines to the wildtype rescue. We next examined the ability of the transgenic proteins to rescue the lethality normally seen in *syt1* null mutants (*Figure 3—figure supplement 2D*). The wildtype Syt1 transgene robustly rescued lethality, with the R250H and K379E transgenes also generating very significant rescue. In contrast, S332L provided very little rescue, while R334H and P363S failed to rescue

the lethality. The lack of the ability of the S332L, R334H and P363S mutant Syt1 proteins to provide significant behavioral rescue argues that these mutations are more likely to significantly disrupt the properties of Syt1 in vivo than R250H and K379E.

We first characterized the effects of the C2A R250H mutation that we predicted might alter C2A-C2B domain interactions and Syt1 multimerization. The R250H Syt1 transgene rescued several aspects of the null phenotype, indicating interactions regulated by this residue are not absolutely essential for Syt1 function, consistent with the behavioral rescue we observed. However, evoked neurotransmitter release was decreased by 60% compared to WT Syt1 transgenic rescues (*Figure 3A,B*). While fusion is largely asynchronous in *syt1* null mutants, release in the R250H mutant was synchronous with little asynchronous fusion observed (*Figure 3C–F*). The R250H mutation had a surprising effect on spontaneous release, with a significant increase in mini frequency compared to controls and null mutants (*Figure 3G*, *Figure 3—figure supplement 3*). As such, interactions mediated by R250 play a role in clamping spontaneous release, but do not participate in suppressing the enhanced asynchronous fusion events observed in *syt1* nulls. The divergent effects on suppression of spontaneous versus asynchronous release indicate that independent interactions regulate these inhibitory properties of Syt1. We conclude that R250-mediated interactions are important for triggering normal amounts of vesicle fusion during an evoked response, but are not as critical for the timing of fusion.

## Molecular modeling indicates the R250H mutation destabilizes C2A-C2B inter-domain stability and the Syt1 dimer

To determine how R250H may disrupt Syt1 function, we performed molecular modeling to examine how alterations in this residue might change C2A-C2B domain interactions and Syt1 dimerization. We used the mammalian Syt1 crystal structure for modeling analysis. The R199 residue, which is homologous to *Drosophila* R250, is positioned at interface of the C2A and C2B domains (*Figure 1J*, *Figure 2A*). We predicted that mutating the positively charged R250 residue at the interface of the two rigid C2 domains might induce a conformational rearrangement. To explore this possibility, we performed prolonged molecular dynamics (MD) simulations of native and R199H mammalian Syt1 (*Figure 4*). We first investigated the stability of the native Syt1 conformational state observed by crystallography by performing MD simulations at the microsecond scale in a water/ion environment. The MD trajectory (*Figure 4A–D*) demonstrated that the domains remain relatively stable, though minor conformational transitions were observed. More specifically, the inter-domain organization remained stable for approximately 1 µs (*Figure 4D*, states 1–3), with only minor and transient separations between the C2 domains (*Figure 4D*, state 1; note minor fluctuations in the root mean-square deviation (RMSD) and gyration radius around the 0.5 µs time point, *Figure 4A–C*). This was followed by a period of instability (transient raise in RMSD around 1.2–1.5 µs, *Figure 4A,B*). This instability was associated with a separation of the C2 domains (transient increase in gyration radius, *Figure 4C*) and inter-domain rotation (*Figure 4D*, state 3). Subsequently, the structure transitioned to a different state with tightly interacting C2 domains (*Figure 4D*, state 4), which was stable for the remainder of the observed trajectory (*Figure 4A–C*, 1.8–3.3 µs). This conformation was fairly similar to initial state 0: the C2A and C2B domains were perpendicularly oriented and interacted with the $Ca^{2+}$ binding loops of the C2A domain, forming contacts with a C2B alpha helix. These results suggest that although Syt1 is likely to undergo minor conformational rearrangements in solution, the organization observed by crystallography with perpendicularly oriented and tightly interacting domains is likely to be stable. Interestingly, in the final state of the Syt1 trajectory, R199 is positioned at the interface of the tightly interacting C2A and C2B domains (*Figure 4E*).

We next explored how the R199H mutation would affect the dynamics of the Syt1 C2 domains (*Figure 4F–I*). Strikingly, the mutation drastically destabilized the Syt1 inter-domain organization. After a brief period of stability (0.15 us), the domains separated (note a jump in RMSD in *Figure 4F and a* peak in gyration radius in *Figure 4H*), and multiple conformational transitions followed (*Figure 4I*). The entire conformational dynamics (*Figure 4F–L*), including the overall RMSD (*Figure 4F,J*) and the extent of the conformational rearrangements (*Figure 4G,K*), was drastically affected by mutation of R199. Importantly, the mutated Syt1 structure had an increased gyration radius (*Figure 4H,L*), indicating a larger separation of the C2 domains likely causes the conformational transitions. This structural flexibility contrasts with native Syt1 dynamics, where domains interact tightly and conformational transitions are minor and infrequent (*Figure 4J–L*).

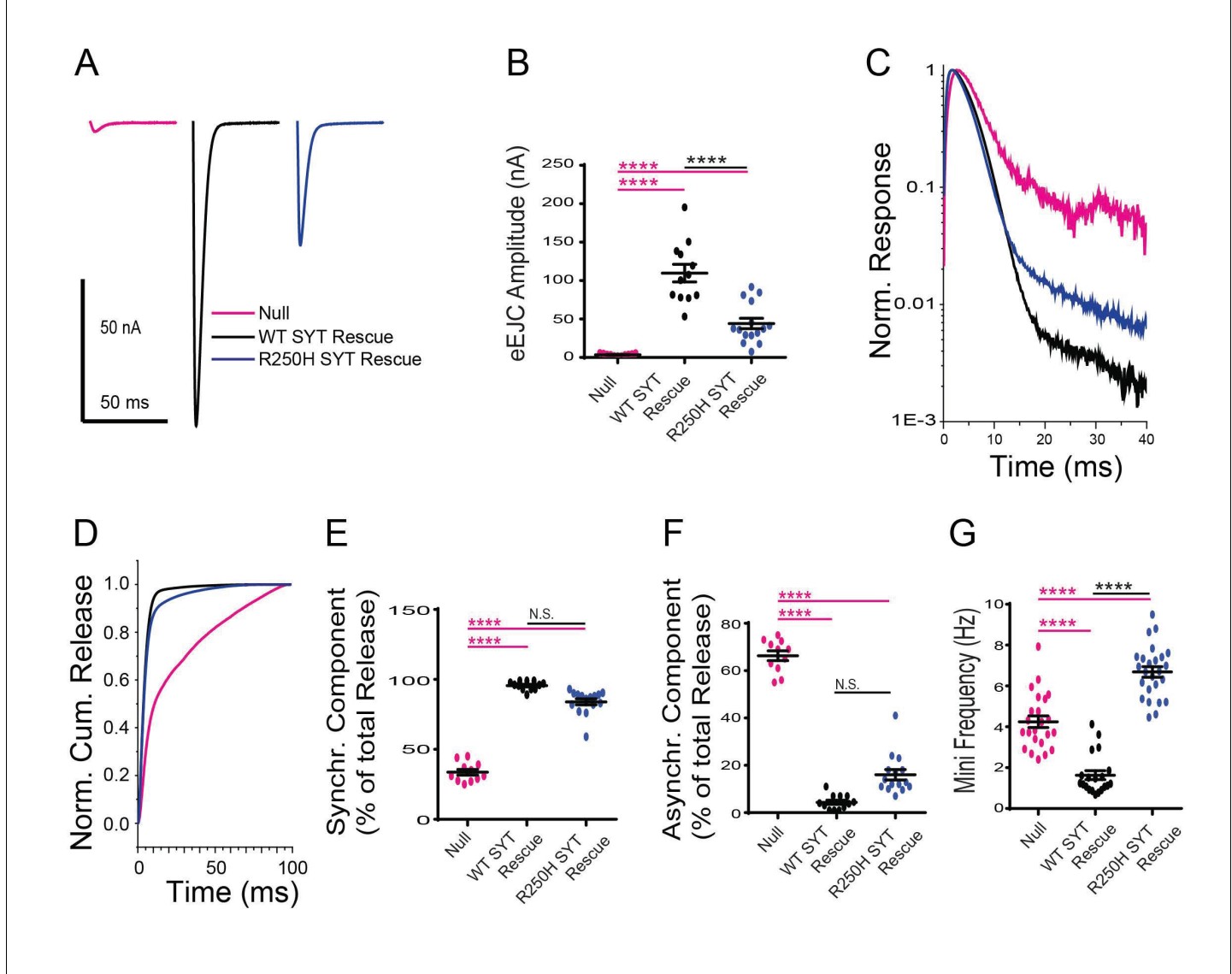

**Figure 3.** Effects of the R250H mutation on synaptic vesicle fusion. (**A**) Representative eEJCs recorded in 2.0 mM extracellular Ca$^{2+}$ in *syt1*$^{-/-}$ null larvae (magenta), and null mutants rescued with wildtype Syt1 (WT SYT rescue, black) or R250H Syt1 (R250H SYT rescue, blue). (**B**) Quantification of mean eEJC amplitudes in the indicated genotypes: null, 3.5 ± 0.4 nA, n = 11; WT Syt1 rescue, 109.7 ± 11.5 nA, n = 12; R250H Syt1 rescue, 44.1 ± 6.8 nA, n = 15. (**C**) Average normalized responses for each genotype plotted on a semi-logarithmic graph to display release components. (**D**) Cumulative release normalized for the maximum in 2.0 mM Ca$^{2+}$ for each genotype. Each trace was adjusted to a double exponential fit. (**E**) Quantification of the synchronous component of release for each genotype: null, 33.7 ± 2.1%, n = 11; WT Syt1 rescue, 95.6 ± 0.9%, n = 12; R250H Syt1 rescue, 84.0 ± 2.2%, n = 15. (**F**) Quantification of the asynchronous components of release for each genotype: null, 66.3 ± 2.1%, n = 11; WT Syt1 rescue, 4.4 ± 0.9%, n = 12; R250H Syt1 rescue, 16.0 ± 2.2%, n = 15. (**G**) Quantification of average mini frequency for the indicated genotypes: null, 4.2 ± 0.3 Hz, n = 23; WT Syt1 rescue, 1.6 ± 0.2 Hz, n = 20; R250H Syt1 rescue, 6.7 ± 0.3 Hz, n = 25. Statistical significance was determined using one-way ANOVA (nonparametric) with post hoc Sidak's multiple comparisons test. N.S. = no significant change, ****p<0.0001. Error bars are SEM.

DOI: https://doi.org/10.7554/eLife.28409.012

The following source data and figure supplements are available for figure 3:

**Source data 1.** Sample size (n), mean, SEM, and One-way Anova (and nonparametric) Sidak's multiple comparisons test are presented for the data in *Figure 3B*, *Figure 6B* and *Figure 8B*.

DOI: https://doi.org/10.7554/eLife.28409.016

**Source data 2.** Sample size (n), mean, SEM, and One-way Anova (and nonparametric) Sidak's multiple comparisons test are presented for the data in *Figure 3E*, *Figure 6E* and *Figure 8E*.

DOI: https://doi.org/10.7554/eLife.28409.017

*Figure 3 continued on next page*

*Figure 3 continued*

**Source data 3.** Sample size (n), mean, SEM, and One-way Anova (and nonparametric) Sidak's multiple comparisons test are presented for the data in *Figure 3F*, *Figure 6F* and *Figure 8F*.

DOI: https://doi.org/10.7554/eLife.28409.018

**Source data 4.** Sample size (n), mean, SEM, and One-way Anova (and nonparametric) Sidak's multiple comparisons test are presented for the data in *Figure 3G*, *Figure 6G* and *Figure 8G*.

DOI: https://doi.org/10.7554/eLife.28409.019

**Source data 5.** Sample size (n), mean, SEM, and One-way Anova (and nonparametric) multiple comparisons test are presented for the data in *Figure 3—figure supplement 2*

DOI: https://doi.org/10.7554/eLife.28409.020

**Figure supplement 1.** Syt1 mutant proteins normally target to the synapse.

DOI: https://doi.org/10.7554/eLife.28409.013

**Figure supplement 2.** Comparison of wildtype (white) animals and Syt1 transgenic rescue.

DOI: https://doi.org/10.7554/eLife.28409.014

**Figure supplement 3.** Postsynaptic current recordings of spontaneous release at third instar larval muscle 6 NMJs of the indicated genotypes in HL3.1 saline with 2 mM Ca$^{2+}$.

DOI: https://doi.org/10.7554/eLife.28409.015

Multiple studies suggest Syt1 multimerization may be important in vivo. As such, we next explored the stability of the Syt1 dimer in solution, how it might assemble at the interface of the synaptic vesicle and plasma membrane, and how conformational dynamics of a Syt1 dimer would be affected by the R199H mutation. Since the Syt1 structure obtained by crystallography represented a dimer (*Fuson et al., 2007*), we used the dimer as an initial approximation for energy minimization followed by a 2.8 μs MD simulation in a water/ion environment. We found the dimer remained stable and underwent only subtle rearrangements of the loop regions, but inter-domain and monomer rearrangement did not occur (*Figure 5A*), and no major conformational transitions were observed. The dimer was stabilized by hydrophobic interactions of R199, which formed a salt bridge with N207 and tight hydrophobic contacts with V205 (*Figure 5B*). The MD stimulation of the Syt1 dimer suggest the structure observed by crystallography is likely to be stable in solution and not an artifact of crystallization.

To explore how a Syt1 dimer might interface between the two membranes involved in fusion, we created a model of the dimer on the lipid bilayer surface by positioning it on a plasma membrane bilayer and performing energy minimization (*Figure 5C*). Notably, the C2B polybasic stretch from residues 321 to 327 (KELKKKK) of both Syt1 monomers engaged the lipid bilayer, positioning the C2B Ca$^{2+}$ binding loops oriented towards the plasma membrane (*Figure 5C*). In contrast, the Ca$^{2+}$ binding loops of the C2A domains emerged from the opposite side of the structure, indicating they may interact with the synaptic vesicle lipid bilayer rather than the plasma membrane. Such an arrangement suggests a model whereby the Syt1 dimer could act as an intermediate in the fusion pathway. We hypothesize that the Syt1 dimer could stabilize a state with the synaptic vesicle and the plasma membrane in close proximity, but not merging, thus inhibiting spontaneous fusion. Ca$^{2+}$ binding would then induce penetration of the C2B domains into the plasma membrane and penetration of the C2A domains into the vesicle membrane, bridging the two membranes to create tension and promote stalk formation and fusion pore opening.

The central site of R199 at the Syt1 dimer interface (*Figure 5B*) suggests that mutating this residue could destabilize dimer formation. To test this prediction directly, we performed MD simulations of the mutated Syt1 dimer and observed that dimer structure and organization was substantially disrupted (*Figure 5D*). Within the initial 0.5 μs of the MD trajectory (*Figure 5D*), both monomers underwent conformational transitions associated with inter-domain rotation, and subsequently the structure transitioned into a state where the symmetric organization of the monomers was disrupted and the Ca$^{2+}$ binding loops of the C2A domains faced different surfaces of the dimer (*Figure 5D*, 1.7 μs time point). This transition is illustrated by the major RMSD deviation of the mutated dimer from its initial state (*Figure 5D*). This result suggests that although dimerization may still occur in the R250H mutant background, the dimer structure observed for native Syt1 would be less stable.

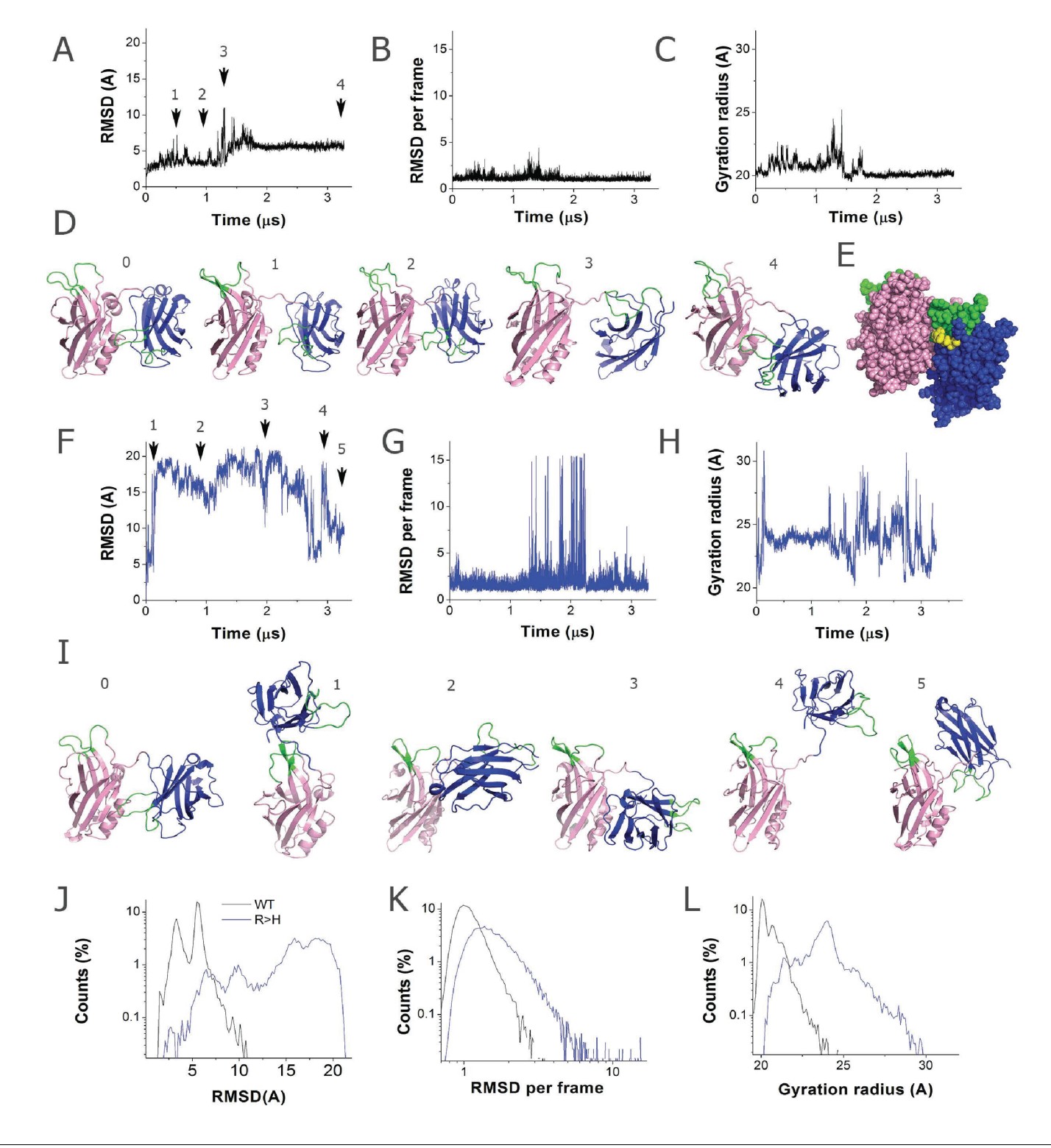

**Figure 4.** The R199H mutation (homologous to *Drosophila* R250H) drastically enhances Syt1 conformational flexibility. (**A**) RMSD computed for backbone atoms of mammalian Syt1 along the MD trajectory. Note a region of instability between 0.5 and 1 µs, which is followed by a conformational transition. Arrows indicate the states shown in D. (**B**) Backbone RMSD per frame along the trajectory. Note a region of instability in the middle of the trajectory, followed by a relatively stable state (state 4, panel D) between 1.8 and 3.3 µs of the simulation. (**C**) Gyration radius along the trajectory indicating the separation between domains. Note a transient separation of the domains at 1.2–1.3 µs, which is followed by a decrease in gyration

*Figure 4 continued on next page*

*Figure 4 continued*

radius, corresponding to tightening of the overall structure. (D) Subsequent states of Syt1 along the trajectory. Note that the organization of the domains remains stable from state 0 to state 2, with a light separation of the domains in state 1. The transient state 3 has separated domains with the $Ca^{2+}$-binding loops facing the same surface. This state transitions to a stable state 4 with $Ca^{2+}$ binding loops of the C2A domain interacting with the alpha-helix of the C2B domain, similar to the initial state 0. (E) In the final state of the Syt1 trajectory, residue R199 (yellow) is positioned at the interface of the C2A and C2B domains. (F) RMSD for R199H Syt1 shows a major conformational transition in the beginning of the trajectory, as well as a lack of overall stability. (G) RMSD for R199H Syt1 shows multiple conformational transitions, especially in the second half of the trajectory. (H) The gyration radius of R199H Syt1 shows a separation of the domains in the beginning of the trajectory, followed by major fluctuations, corresponding to transient separation and tightening of the domains. (I) Subsequent states of R199H Syt1 along the trajectory show major rearrangements of the domains. (J) The trajectory of R199H Syt1 has an RMSD distribution shifted to the right, indicating more prominent conformational transitions. Note two closely positioned peaks in the WT distribution, corresponding to two stable conformational states along the trajectory. In contrast, the trajectory of mutated Syt1 shows five RMSD peaks, corresponding to five different conformations, all shifted to the right and considerably deviated from the initial state. (K) The RMSD per frame distribution is shifted for R199H Syt1, demonstrating more prominent conformational changes from frame to frame. (L) The gyration radius distribution is shifted in mutant Syt1, demonstrating a larger separation between the domains.

DOI: https://doi.org/10.7554/eLife.28409.021

## Characterization of the role of K379 and P363 in synaptic vesicle fusion

We next examined how the K379E mutation alters Syt1 function. K379 resides on the C2B domain polybasic strand and has been suggested to dock Syt1 onto the plasma membrane in a $Ca^{2+}$-independent manner. We generated the K379E mutation in the Syt1 wildtype transgene and expressed it in the null background. In contrast to the robust ability of the K379E mutation to disrupt the dominant-negative effects of the C2B $Ca^{2+}$ binding mutants on fusion, the mutation had only modest effects on wildtype Syt1 activity and was able to significantly rescue the lethality observed in the *syt1* null mutant. Release in the K379E mutation background was synchronous, with little asynchronous release observed (*Figure 6A–F*). The primary defect identified was a ~ 50% reduction in the evoked EJC amplitude (*Figure 6A,B*), indicating K379 regulates the total amount of vesicles that fuse, but does not play a critical role in release timing or the suppression of asynchronous release. The K379E mutation did not rescue the enhanced spontaneous neurotransmitter release observed in the null mutant (*Figure 6G*). We conclude that the polybasic motif in the C2B domain promotes normal levels of synaptic vesicle release, but is not essential for the process and does not play an important role in suppressing asynchronous fusion.

The P363S mutation is located near the C2B $Ca^{2+}$ binding D2 site (D362), suggesting it may disrupt the ability of the C2B domain to insert into lipid bilayers in a $Ca^{2+}$-dependent manner. We generated the P363S mutation in the Syt1 wildtype transgene and expressed the construct in the *syt1* null background. The P363S mutation resulted in a severe reduction in evoked EJC amplitude (*Figure 6A,B*), as well as altering synchronous versus asynchronous components of release (*Figure 6C–F*). Similar to the null mutant, the amount of asynchronous release was dramatically increased (*Figure 6C,D,F*). These observations are consistent with previous results indicating the C2B $Ca^{2+}$ binding domain is an essential trigger for synaptic vesicle exocytosis. The P363S transgene largely rescued the enhanced spontaneous release observed in the null mutant (*Figure 6E*), suggesting this residue does not play a major role in helping Syt1 function as a clamp for spontaneous release.

## Residues S332 and R334 play essential roles in triggering synchronous fusion and suppressing asynchronous release

We next examined how mutations on the C2B domain surface opposite the polybasic stretch alter Syt1 function. Resides S332 (corresponding to mammalian S279) and R334 (corresponding to mammalian R281) decorate the primary interface of the recently identified Syt1-SNARE binding surface (*Zhou et al., 2015*) (*Figure 7A,B*). Residue R334 has a prominent role in the Syt1-SNARE complex by forming a salt bridge with residue E55 of SNAP25, while S332 participates in weaker hydrophobic SNAP-25 interactions (*Figure 7C,D*). As such, these mutations would be predicted to disrupt SNARE-Syt1 interactions in vivo. We generated S332L and R334H mutations in the wild type Syt1 transgene and expressed them in the *syt1* null background (*Figure 8*). Compared to control rescues, mutations at these sites severely disrupted the ability of Syt1 to support synchronous exocytosis (*Figure 8A,B*) and failed to rescue most Syt1-dependent functions. The R334H mutation had a more

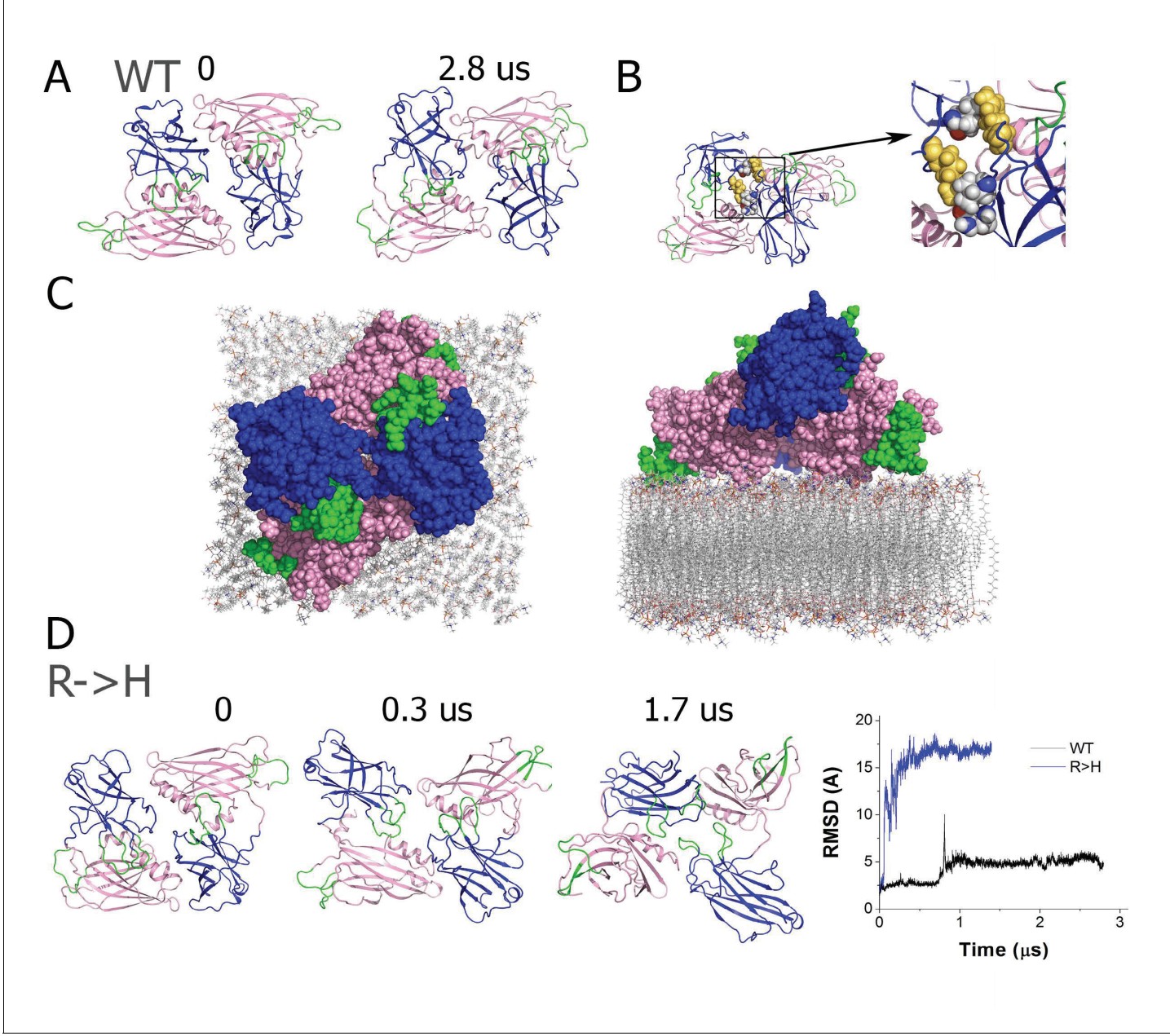

**Figure 5.** The R199H mutation destabilizes the Syt1 dimer. (**A**) The dimer structure of native Syt1 is stable in the water/ion environment at the microsecond time scale. Note similar arrangements of monomers and dimers in the initial and final states. (**B**) The R199 residue (yellow) is central in van der Waals interactions between Syt1 monomers that form the dimer structure. (**C**) Structural model of the Syt1 dimer on top of the plasma membrane from a top-down view (left) and a perpendicular view to the membrane (right). Note the polybasic region of the C2B domain (magenta) interacts with the membrane, positioning the C2B $Ca^{2+}$ binding loops (green) near the plasma membrane. In contrast, the $Ca^{2+}$ binding loops (green) of the C2A domain (blue) face the opposite side of the structure, and could serve to interact with the synaptic vesicle membrane to bridge the two bilayers. (**D**) The dimeric structure of Syt1 is disrupted in the R199H mutant. Three states along the trajectory of the mutated dimer show a major rearrangement of the domains. The RMSD plot on the right reveals a major conformational transition at the beginning of the trajectory of R199H Syt1 (blue trace) compared to the trajectory of native Syt1 (black trace).

DOI: https://doi.org/10.7554/eLife.28409.022

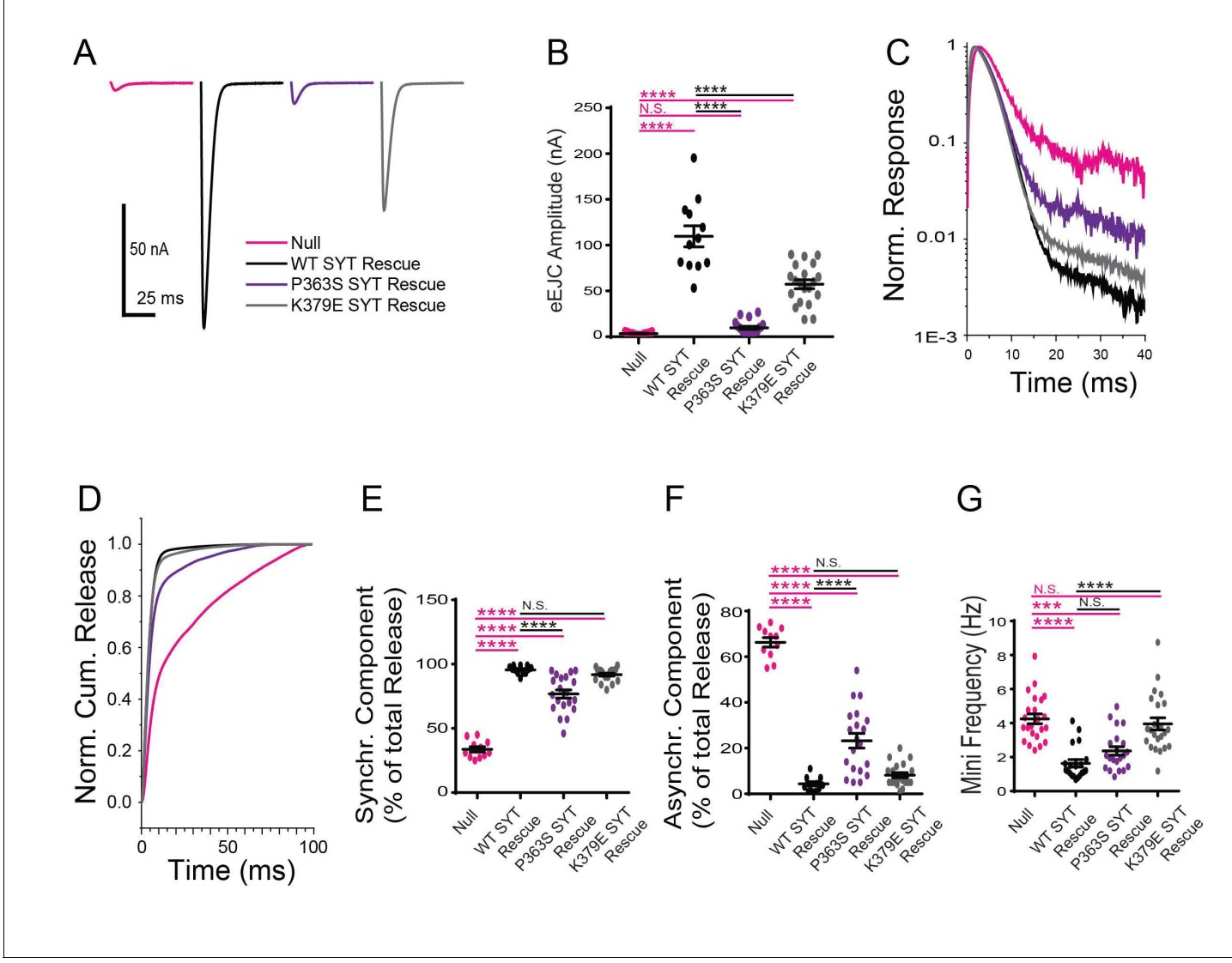

**Figure 6.** Physiological analysis of the K379E and P363S mutants on neurotransmitter release. (**A**) Representative eEJCs recorded in 2.0 mM extracellular $Ca^{2+}$ in *syt1* $^{-/-}$ null larvae (magenta) and null mutants rescued with WT Syt1 (black), P363S Syt1 (purple) or K379E Syt1 (grey). (**B**) Quantification of mean eEJC amplitude for the indicated genotypes: null, 3.5 ± 0.4 nA, n = 11; WT Syt1 rescue, 109.7 ± 11.5 nA, n = 12; P363S Syt1 rescue, 9.8 ± 1.7 nA, n = 20; K379E Syt1 rescue, 57.3 ± 4.9 nA, n = 20. C. Average normalized responses plotted on a semi-logarithmic graph to display their components. (**C**) Average normalized responses for each genotype plotted on a semi-logarithmic graph to display release components. (**D**) Cumulative release normalized for the maximum in 2.0 mM $Ca^{2+}$ for each genotype. Each trace was adjusted to a double exponential fit. (**E**) Quantification of the synchronous components of release for each genotype: null, 33.7 ± 2.1%, n = 11; WT Syt1 rescue, 95.6 ± 0.9%, n = 12; P363S Syt1 rescue, 76.8 ± 3.2%, n = 20; K379E Syt1 rescue, 91.8 ± 1.1%, n = 20. (**F**) Quantification of the asynchronous components of release for each genotype: null, 66.3 ± 2.1%, n = 11; WT Syt1 rescue, 4.4 ± 0.9%, n = 12; P363S Syt1 rescue, 23.2 ± 3.2%, n = 20; K379E Syt1 rescue, 8.2 ± 1.1%, n = 20. (**G**) Quantification of average mini frequency for the indicated genotypes: null, 4.2 ± 0.3 Hz, n = 23; WT Syt1 rescue, 1.6 ± 0.2 Hz, n = 20; P363S Syt1 rescue, 2.4 ± 0.2 Hz, n = 21; K379E Syt1 rescue, 4.0 ± 0.3 Hz, n = 23. Statistical significance was determined using one-way ANOVA (nonparametric) with post hoc Sidak's multiple comparisons test. N.S. = no significant change, ***p<0.0005, ****p<0.0001. All error bars are SEM.
DOI: https://doi.org/10.7554/eLife.28409.023

severe phenotype compared to S332L, consistent with the location of R334 at the center of the Syt1-SNARE interaction surface. In contrast to the effects of the R250H and K379E mutations on release, S332L and R334H rescue lines failed to suppress asynchronous release, similar to what is observed in the *syt1* null (***Figure 8C–F***). As such, Syt1-SNARE interactions are likely to be essential for both driving synchronous fusion and preventing activation of release through the asynchronous

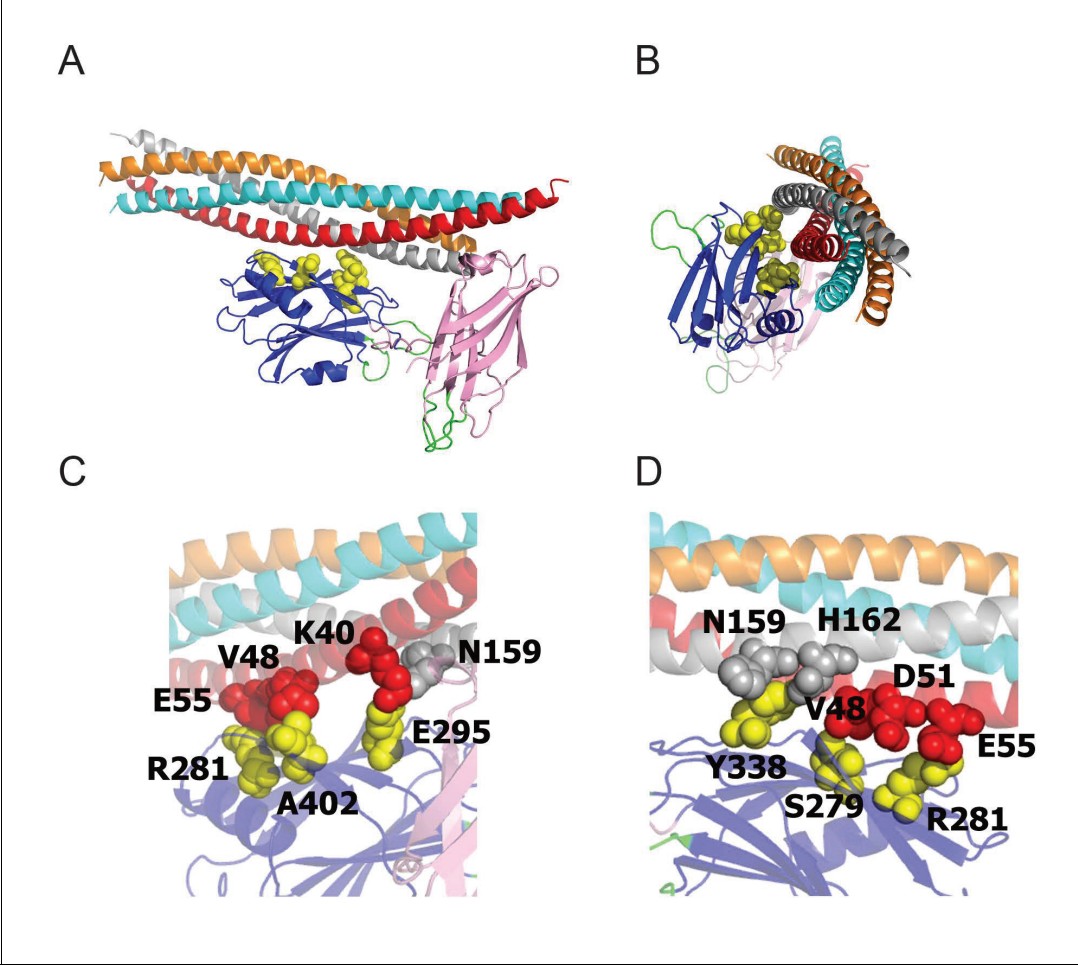

**Figure 7.** The Syt1-SNARE interaction surface (*Zhou et al., 2015*) reveals that residues S332 (S279 in mammalian Syt1), R334 (R281), Y391 (Y338), E348 (E295), and A455 (A402) are positioned at the Syt1-SNARE interface. All five residues were identified as suppressors in our genetic screen. (**A,B**) Two perpendicular views of the Syt1-SNARE complex are shown. The residues listed above are shown in yellow. Orange – synaptobrevin; cyan – syntaxin; red – SN1 domain of SNAP25; silver – SN2 domain of SNAP25. (**C,D**) Magnified views of opposite surfaces of the Syt1-SNARE interface are shown. Note that the Syt1-SNARE interaction is supported by a salt bridge between R334 (R281) of Syt1 and E55 of SNAP25. In addition, S332 (S279) forms hydrophobic interactions with V48 of SNAP25.

DOI: https://doi.org/10.7554/eLife.28409.024

pathway. In addition to their effects on the amount and timing of evoked release, both mutations failed to suppress enhanced spontaneous release (*Figure 8G*). Notably, mutations in R334 had an even larger increase in mini frequency than the null mutant, indicating Syt1-SNARE interactions are critical for preventing spontaneous exocytosis of synaptic vesicles in the absence of $Ca^{2+}$ influx.

## Examining $Ca^{2+}$ cooperativity and synaptic vesicle distribution in the Syt1 suppressor lines

In addition to regulating synchronous and asynchronous release, binding of $Ca^{2+}$ by Syt1 is a key component of the higher order $Ca^{2+}$ cooperativity (n = ~3–5 depending on synapse) observed for synaptic vesicle fusion (*Kaeser and Regehr, 2014*; *Littleton et al., 1994*; *Yoshihara and Littleton, 2002*). In addition, Syt1 participates in synaptic vesicle docking (*Reist et al., 1998*) and endocytosis (*Littleton et al., 2001*; *Poskanzer et al., 2006*; *2003*). To examine the effects of the suppressor mutations on these synaptic parameters, we examined $Ca^{2+}$ cooperativity of release (*Figure 8H*) and performed EM analysis (*Figure 9*) on the transgenic rescue lines for several of the point mutants. For $Ca^{2+}$ cooperativity measurements, we assayed the evoked EJC amplitude in 10 different extracellular $Ca^{2+}$ concentrations ranging from 0.1 mM to 4 mM. $Ca^{2+}$ cooperativity in control animals (WT SYT

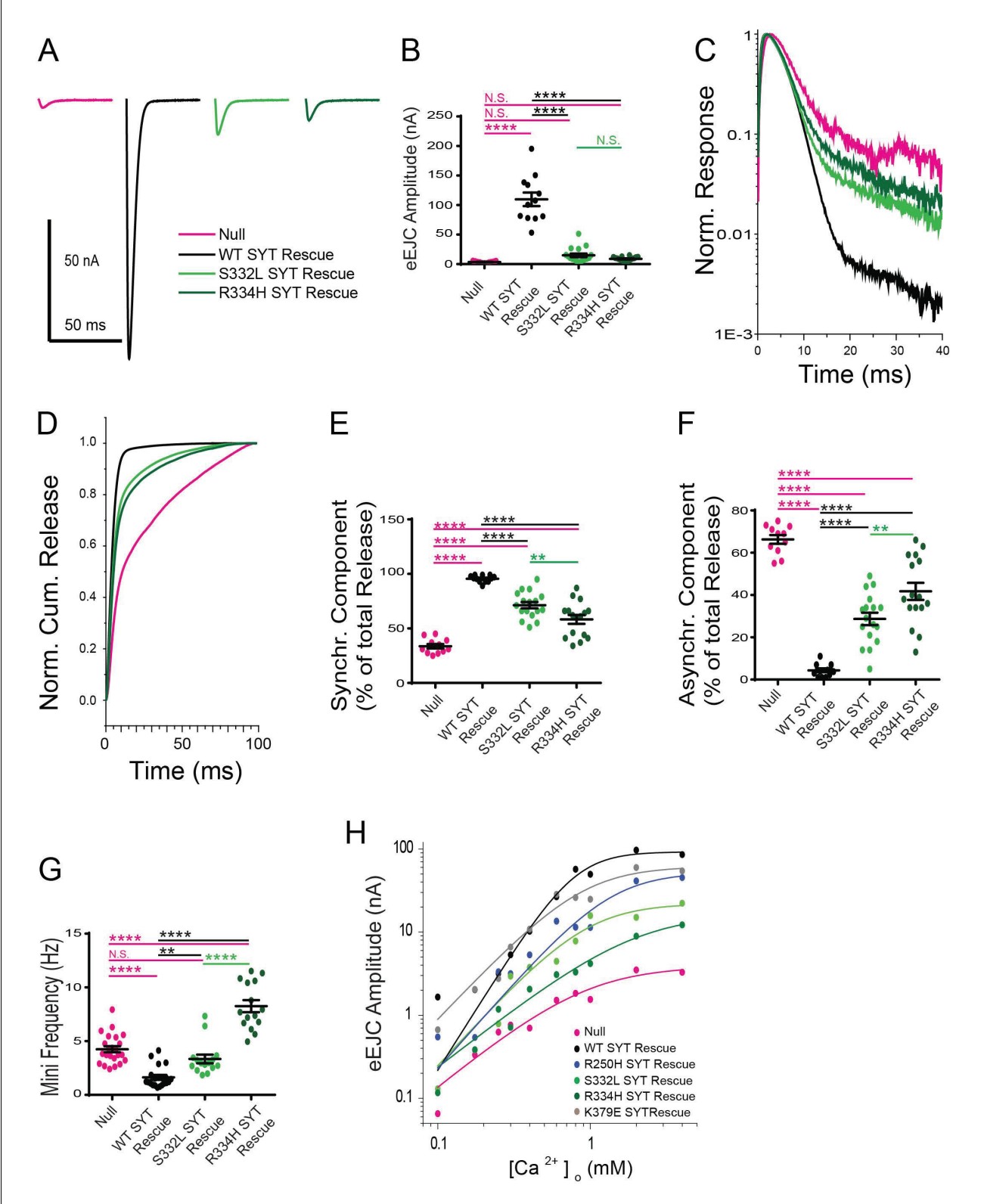

**Figure 8.** Effects of the S332L and R334H mutations on neurotransmitter release. (A) Representative eEJCs recorded in 2.0 mM extracellular Ca$^{2+}$ in *syt1* $^{-/-}$ null larvae (magenta) and null mutants rescued with WT Syt1 (black), S332L Syt1 (light green) or R334H Syt1 (dark green). (B) Quantification of mean eEJC amplitudes for the indicated genotypes: null, 3.5 ± 0.4 nA, n = 11; WT Syt1 rescue, 109.7 ± 11.5 nA, n = 12; S332L Syt1 rescue, 14.9 ± 3.0 nA, n = 17; R334H Syt1 rescue, 8.8 ± 0.9 nA, n = 16. (C) Average normalized responses plotted on a semi-logarithmic graph to display their components. (D)

*Figure 8 continued on next page*

*Figure 8 continued*

Cumulative release normalized for the maximum in 2.0 mM $Ca^{2+}$ for each genotype. Each trace was adjusted to a double exponential fit. (E) Quantification of the synchronous components of release for each genotype: null, 33.7 ± 2.1%, n = 11; WT Syt1 rescue, 95.6 ± 0.9%, n = 12; S332L Syt1 rescue, 71.3 ± 2.9%, n = 17; R334H Syt1 rescue, 58.3 ± 4.0%, n = 16. (F). Quantification of the asynchronous components of release for each genotype: null, 66.3 ± 2.1%, n = 11; WT Syt1 rescue, 4.4 ± 0.9%, n = 12; S332L Syt1 rescue, 28.7 ± 2.9%, n = 17; R334H Syt1 rescue, 41.8 ± 4.0%, n = 16. (G) Quantification of average mini frequency for the indicated genotypes: null, 4.2 ± 0.3 Hz, n = 23; WT Syt1 rescue, 1.6 ± 0.2 Hz, n = 20; S332L Syt1 rescue, 3.3 ± 0.4 Hz, n = 15; R334H Syt1 rescue, 8.3 ± 0.5 Hz, n = 15). For panels B-G, statistical significance was determined using one-way ANOVA (nonparametric) with post hoc Sidak's multiple comparisons test. N.S. = no significant change, **=p < 0.01, ****=p < 0.0001. All error bars are SEM. (H) $Ca^{2+}$ cooperativity of release is shown on a double logarithmic plot, with Hill fit for each genotype. Ten extracellular $Ca^{2+}$ concentrations (mM) were tested: 0.1, 0.175, 0.25, 0.3, 0.4, 0.6, 0.8, 1, 2, 4. The cooperativity for the genotypes is (value ±standard error): null, 1.5 ± 0.4 (n = 102); WT Syt1 rescue, 2.9 ± 0.7 (n = 119); R250H Syt1 rescue, 2.1 ± 0.6 (n = 112); S332L Syt1 rescue, 2.0 ± 0.6 (n = 112); R334H Syt1 rescue, 1.4 ± 0.2 (n = 112); K379E Syt1 rescue, 1.9 ± 0.6 (n = 118). Regression analysis revealed significant differences in the WT Syt1 rescue (p<0.0001) and R334H Syt1 rescue (p<0.05) compared to null. The R250H Syt1 rescue (p<0.05) and the S332L Syt1 rescue (p<0.0001) were significantly different compared to WT Syt1 rescue. At least 6 recordings were performed in each concentration, with n indicating the total number of recordings in the 10 different $Ca^{2+}$ concentrations.
DOI: https://doi.org/10.7554/eLife.28409.025

The following source data is available for figure 8:

**Source data 1.** Sample size (n), mean, SEM are presented for the data in *Figure 8H*.
DOI: https://doi.org/10.7554/eLife.28409.026
**Source data 2.** Sample size (n), mean, SEM, and Hill Fit of Null are presented for the data in *Figure 8H*.
DOI: https://doi.org/10.7554/eLife.28409.027
**Source data 3.** Sample size (n), mean, SEM, and Hill Fit of WT SYT Rescue are presented for the data in *Figure 8H*.
DOI: https://doi.org/10.7554/eLife.28409.028
**Source data 4.** Sample size (n), mean, SEM, and Hill Fit of R250H SYT Rescue are presented for the data in *Figure 8H*.
DOI: https://doi.org/10.7554/eLife.28409.029
**Source data 5.** Sample size (n), mean, SEM, and Hill Fit of K379E SYT Rescue are presented for the data in *Figure 8H*.
DOI: https://doi.org/10.7554/eLife.28409.030
**Source data 6.** Sample size (n), mean, SEM, and Hill Fit of S332L SYT Rescue are presented for the data in *Figure 8H*.
DOI: https://doi.org/10.7554/eLife.28409.031
**Source data 7.** Sample size (n), mean, SEM, and Hill Fit of R334H SYT Rescue are presented for the data in *Figure 8H*.
DOI: https://doi.org/10.7554/eLife.28409.032
**Source data 8.** Regression Analysis are presented for the data in *Figure 8H*.
DOI: https://doi.org/10.7554/eLife.28409.033

rescue) was 2.9, compared to 1.5 in *syt1* null mutants. The most dramatic effect on $Ca^{2+}$ cooperativity was observed in the R334H rescue, where the $Ca^{2+}$ dependence of release was reduced to 1.4, similar to the *syt1* null. These data indicate that Syt1-SNARE binding mediated by the R334 residue is a key determinant of the $Ca^{2+}$ cooperativity for synaptic vesicle fusion.

To examine synaptic ultrastructure in the rescue lines, we performed transmission electron microscopy (TEM) on third instar larval NMJs and quantified synaptic vesicle density within boutons (*Figure 9A,C*) and vesicle distribution surrounding active zone T-bars (*Figure 9B*). Two primary defects were observed in *syt1* null mutants compared to wildtype Syt1 transgenic rescue lines. First, there was a 4-fold decrease in the total density of synaptic vesicles within nerve terminals (*Figure 9A,C*), consistent with the role of Syt1 in vesicle endocytosis. In addition, there was a 60% reduction in docked synaptic vesicles directly underneath the T bar (*Figure 9B*). Expression of a wild-type Syt1 transgene in the null mutant background rescued these phenotypes. We quantified vesicle clustering near T-bars and in a radius of 100 nm, 100–150 nm, and 150–200 nm from the T-bar for the R250H, K379E, S332L and R334H mutant rescue lines and compared them to control rescues. In all cases, synaptic vesicle number proportionally increased in areas (100 nm and 150 nm) near the T-bar, but subsided between 150 to 200 nm away from the T-bar, indicating vesicle clustering is most efficient within a 150 nm radius around T-bars. The individual point mutant lines rescued synaptic vesicle docking directly at the active zone, as well as vesicle accumulation within 100 nm of the T-bar (*Figure 9B*). Vesicles were able to cluster around T-bars, even though the total number of synaptic vesicles in the terminal was reduced in several lines (R250H, K379E). As such, the defects in neurotransmitter release observed in these mutants are likely due to post-docking functions for the interactions regulated by each specific residue.

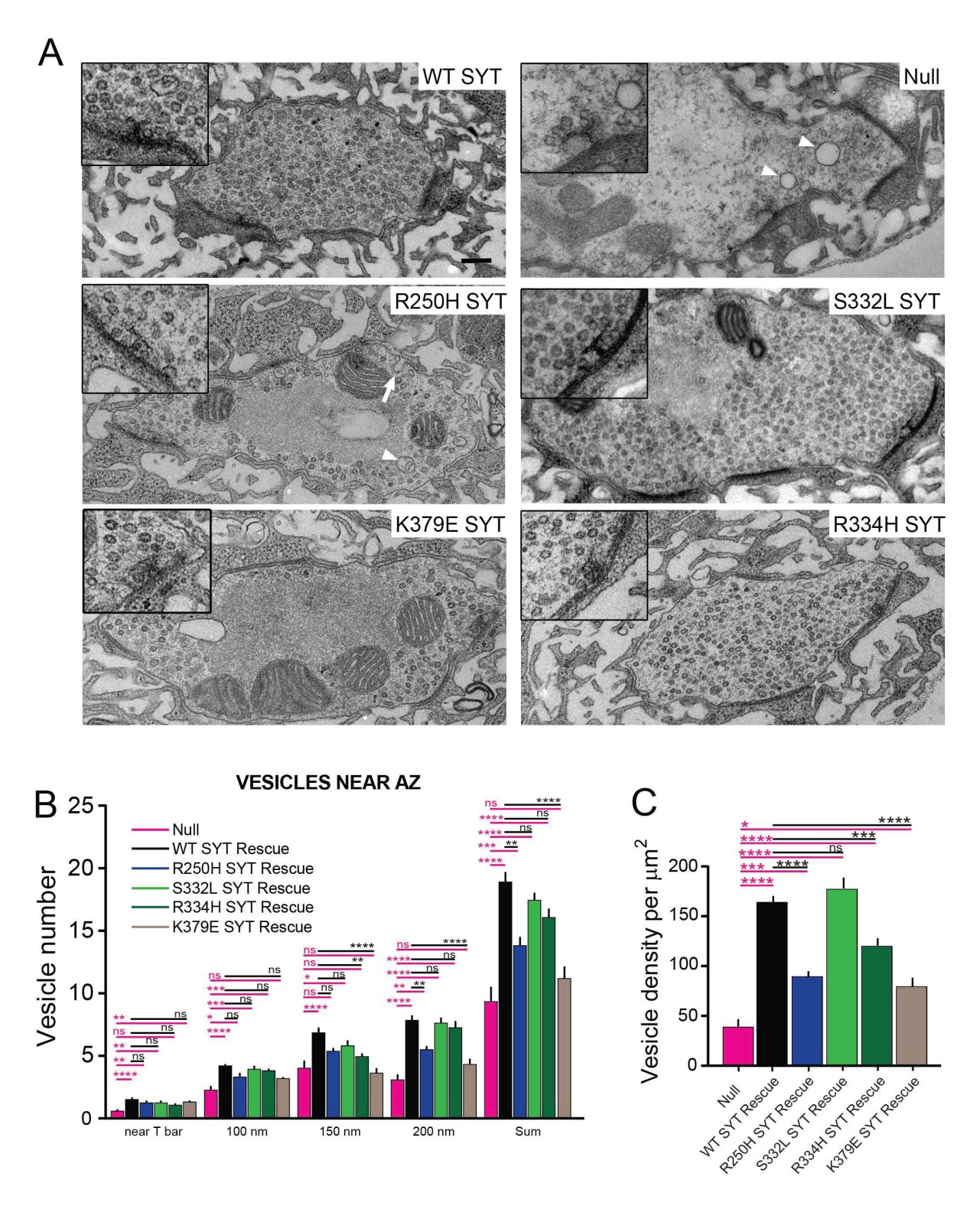

**Figure 9.** Ultrastructural synaptic defects in *syt1* mutations. (**A**) Representative electron micrographs of third instar synaptic boutons from the NMJ of *syt1* null mutants rescued with the indicated transgenic lines. A 2X-magnified view of the T-bar is shown in the insert. Enlarged vacuolar structures prominent in the R250H rescue line are marked by a white arrowhead. Enlarged membrane invaginations in the R250H rescue are indicated by a white arrow. The scale bar is 0.2 µm. (**B**) Quantification of synaptic vesicle number located underneath the T-bar (near T-bar) or within 100 nm, 100–150 nm or

*Figure 9 continued on next page*

*Figure 9 continued*

150–200 nm radius from the T-bar for the indicated genotypes. Quantification of vesicle number near T-bar for each genotype: null, 0.58 ± 0.16, n = 24; WT rescue, 1.5 ± 0.1, n = 20; R250H rescue 1.3 ± 0.1, n = 30; S332L rescue, 1.3 ± 0.1, n = 21; R334H rescue, 1.1 ± 0.1, n = 29; K379E rescue, 1.3 ± 0.2, n = 23. Quantification of vesicle number within 100 nm radius area around T-bar for each genotype: null, 2.3 ± 0.4, n = 24; WT rescue, 4.2 ± 0.2, n = 20; R250H rescue 3.3 ± 0.3, n = 32; S332L rescue, 4.0 ± 0.3, n = 21; R334H Rescue, 3.9 ± 0.2, n = 29; K379E Rescue, 3.2 ± 0.3, n = 23. Quantification of vesicle number within 100–150 nm area around T-bar for each genotype: null, 4.0 ± 0.5, n = 24; WT rescue, 6.9 ± 0.5, n = 20; R250H rescue 5.4 ± 0.3, n = 32; S332L rescue, 4.0 ± 0.3, n = 21; R334H rescue, 5.0 ± 0.3, n = 29; K379E rescue, 3.7 ± 0.4, n = 23. Quantification of vesicle number within 150–200 nm area around T-bar for each genotype: null, 3.2 ± 0.5, n = 24; WT rescue, 7.9 ± 0.5, n = 20; R250H rescue 5.5 ± 0.4, n = 32; S332L rescue, 7.7 ± 0.4, n = 21; R334H rescue, 7.3 ± 0.5, n = 29; K379E rescue, 4.3 ± 0.5, n = 23. Quantification of sum of vesicle number within 200 nm area around T-bar for each genotype: null, 9.3 ± 1.2, n = 24; WT SYT rescue, 18.9 ± 0.9, n = 20; R250H SYT rescue 14.3 ± 0.6, n = 32; S332L SYT rescue, 17.4 ± 0.7, n = 21; R334H SYT rescue, 16.1 ± 0.7, n = 29; K379E SYT rescue, 11.2 ± 0.9, n = 23. (C) Synaptic vesicle number was quantified and normalized to bouton area. *Syt1* null mutants have a prominent reduction in vesicle number compared to control Syt1 rescued lines. Quantification of vesicle density for each genotype: null, 38.7 ± 7.5, n = 14; WT SYT rescue, 164.0 ± 7.4, n = 31; R250H SYT rescue 89.7 ± 6.6, n = 38; S332L SYT rescue, 177.3 ± 11.1, n = 18; R334H SYT rescue, 120.0 ± 6.9, n = 35; K379E SYT rescue, 79.6 ± 8.0, n = 20. Statistical significance was determined using one-way ANOVA (nonparametric) with post hoc Sidak's multiple comparisons test. N.S. = no significant change, *p<0.05, **p≤0.01, ***p≤0.0005, ****p<0.0001. All error bars are standard error of the mean (SEM).

DOI: https://doi.org/10.7554/eLife.28409.034

The following source data and figure supplement are available for figure 9:

**Source data 1.** Sample size (n), mean, SEM, and One-way Anova (and nonparametric) Sidak's multiple comparisons test are presented for the data in *Figure 9B*.
DOI: https://doi.org/10.7554/eLife.28409.036
**Source data 2.** Sample size (n), mean, SEM, and One-way Anova (and nonparametric) Sidak's multiple comparisons test are presented for the data in *Figure 9C*.
DOI: https://doi.org/10.7554/eLife.28409.037
**Source data 3.** Sample size (n), mean, SEM, and One-way Anova (and nonparametric) Sidak's multiple comparisons test are presented for the data in *Figure 9—figure supplement 1*.
DOI: https://doi.org/10.7554/eLife.28409.038
**Figure supplement 1.** Characterization of FM 1–43 endocytosis in R250H and K379E rescue lines.
DOI: https://doi.org/10.7554/eLife.28409.035

Although most of the mutant rescues improved vesicle clustering within 200 nm from T-bar, the K379E strain was only able to rescue vesicle clustering directly at T-bars. In addition, the R250H and K379E mutations showed a strong decrease in total vesicle density (*Figure 9C*). Interestingly, the R250H mutant caused a significant accumulation of large vacuoles with a diameter greater than 100 nm, even greater than that observed in the null mutant (control rescue: 0.74 ± 0.16 vacuoles per μm², n = 31; R250H rescue: 2.9 ± 0.5 vacuoles per μm², n = 38, p<0.001; *syt1* null mutant: 1.64 ± 0.28 vacuoles per μm², n = 16, p<0.01). Other mutations did not display vacuolar accumulation and were not significantly different from control. These vacuolar structures in the R250H mutant were occasionally observed as invaginations from the plasma membrane (*Figure 9A*, white arrow), suggesting a defect in the endocytic pathway that is separable from the vesicle reduction observed in the K379E mutant. Analysis of FM 1–43 dye loading during 10 Hz stimulation also indicate a role for R250 and K379 in the endocytosis process (*Figure 9—figure supplement 1*). Following 1 min of loading, labeled synaptic vesicles are distributed uniformly across the bouton periphery for *syt1* null animals rescued with wildtype Syt1. In contrast, *syt1* null mutants show reduced loading, with small clusters of vesicles distinguishable. The R250H and K379E rescue lines also have diminished loading at 1 min compared to wildtype rescue controls. This defect is more pronounced during shorter stimulations (10 Hz for 30 s), where R250H and K379E rescue lines have reduced loading and show more punctate labeling than controls. To further examine defects caused by R250H and K379E, we used a 10 Hz stimulation paradigm that we previously employed in *complexin* mutants to examine synaptic vesicle pool sizes (*Jorquera et al., 2012*). Vesicles from the readily releasable pool (RRP) are required to drive sustained synaptic responses and are mobilized during stimulation to prevent depletion at active zones. To examine if the pool size was reduced in R250H and K379E mutants, we followed release during 10 Hz stimulation over 1000 stimuli. Both the R250H and the K379E mutant rescues displayed a reduction in in the readily releasable pool and sustained release compared to wildtype rescues (*Figure 9—figure supplement 1C–E*), confirming EM observations of a reduced pool size in these mutants. In summary, these data indicate that the polybasic domain and C2A-C2B

domain organization and/or Syt1 multimerization are directly or indirectly required to facilitate synaptic vesicle endocytosis. In contrast, synaptic vesicle docking at the active zone is likely to require multiple interactions mediated by Syt1 that are not disrupted in the individual point mutants.

## Discussion

Syt1's role as the $Ca^{2+}$ sensor for fast synaptic vesicle exocytosis has been well established, although how it triggers fusion is still being elucidated. A large number of studies have established a role for membrane penetration by the $Ca^{2+}$ binding loops of the C2A and C2B domains of the protein (*Fernández-Chacón et al., 2001*; *Lee et al., 2013*; *Mackler et al., 2002*; *Nishiki and Augustine, 2004*; *Shin et al., 2009*; *Yoshihara et al., 2010*). These interactions with negatively charged lipid bilayers are a hallmark of C2 domain function, and are thought to generate local perturbations in membrane structure that facilitate rapid vesicle fusion. Genetic approaches to specifically mutate the key $Ca^{2+}$-binding residues within the C2 domains have validated the important nature of this interaction in vivo. These studies demonstrated essential roles for the C2B $Ca^{2+}$ lipid binding loops, and important, though less essential roles for the C2A $Ca^{2+}$ binding loops. These approaches relied on targeted mutagenesis of key residues that were first found to be important in vitro based on predicted properties derived from the structure of the protein.

Beyond $Ca^{2+}$-dependent lipid binding, there is still debate about other required interactions mediated through Syt1. The unbiased genetic approach described here identified a set of residues that regulate Syt1 function and suggest functional requirements beyond the $Ca^{2+}$ binding loops. Several key residues were identified that decorate a surface of the Syt1 C2B domain that had been poorly characterized, but recently emerged as a SNARE complex binding surface on Syt1. Remarkably, we identified mutations in residues S332 (S279 in mammalian Syt1), R334 (R281), Y391 (Y338), E348 (E295) and A455 (A402). All five amino acids form the essential surface interaction residues that dock the C2B domain of Syt1 onto the SNARE complex, based on the recently elucidated primary Syt1-SNARE complex structure (*Zhou et al., 2015*). Mutations in the S332 and the R334 residues largely abolished the function of Syt1 in driving synchronous fusion. The R334H mutation, which is predicted to be the most essential residue for SNARE binding, failed to rescue synchronous synaptic vesicle release and prevent the increased asynchronous release that is observed in *syt1* null mutants. These observations match well with mutations in SNAP-25 and Syt1 designed to disrupt this interaction at mammalian synapses (*Schupp et al., 2016*; *Zhou et al., 2015*). In addition, we observed that the R334 mutation abolished the $Ca^{2+}$ cooperativity of release, even though R334 does not reside near the $Ca^{2+}$ binding loops of the protein. Although it is unknown what key features contribute to the $Ca^{2+}$ cooperativity (n = 3 to 5) normally seen for release, the number of $Ca^{2+}$ ions that bind to an individual Syt1 protein or the number of Syt1 proteins on a synaptic vesicle that contribute to fusion have all been considered as possible determinants. In addition, prior studies found defects in $Ca^{2+}$ cooperativity in hypomorphic mutants of the SNARE proteins (*Stewart et al., 2000*). Together with our observations of the R334H mutation, these findings argue that the number of Syt1 molecules bound to SNARE complexes likely contributes to the higher-order $Ca^{2+}$ cooperativity value observed for release.

Although these data indicate the SNARE binding surface of Syt1 is a critical determinant of neurotransmitter release, when this interaction might occur during the fusion process remains to be elucidated. We did not observe a morphological docking defect in the R334H or S332L mutants by electron microscopy, suggesting the Syt1-SNARE interaction is functionally important downstream of synaptic vesicle docking. It is unclear when full assembly of SNARE complexes occurs during the synaptic vesicle cycle. One model suggests full SNARE assembly would trigger bilayer fusion, such that docked vesicles likely contain a partially zippered SNARE complex stabilized by the SNARE-binding protein Complexin. Based on the structure of the Syt1-SNARE complex, it is possible that Syt1 could interact with a partially zippered SNARE complex that contains Complexin. This would allow the Syt1-SNARE interaction to play a role in orienting the Syt1 C2 domains near the site of future membrane interactions, serving as a scaffold for the fusion process. Beyond the defect in synchronous release, the R334H mutant also fails to clamp asynchronous release and the enhanced spontaneous release observed in *syt1* null mutants. As such, a Syt1-SNARE complex interaction after docking but before fusion would allow the complex to be stabilized and primed for Syt1-membrane interactions triggered by $Ca^{2+}$. Syt1 could also interact with the plasma membrane in such a scenario

via its polybasic C2B stretch, which lies on the opposite side of the SNARE binding surface (*Figure 5*). This complex could reduce spontaneous and asynchronous release, with Syt1 sandwiched between the plasma membrane and the SNARE complex. $Ca^{2+}$-triggered lipid binding would be predicted to rotate the Syt1 C2B domain downward and towards the membrane, facilitating full SNARE zippering and rapid fusion.

It is interesting to note that the core residues in Syt1 essential for SNARE binding are not conserved in Syt7, a homolog that has been implicated as the asynchronous $Ca^{2+}$ sensor for synaptic vesicle exocytosis (*Bacaj et al., 2013*; *Jackman et al., 2016*; *Liu et al., 2014*; *Wen et al., 2010*). Of the five key residues we identified on the Syt1-SNARE binding surface (S332, R334, E348, Y391 and A455), only S332 is conserved in Syt7. More surprisingly, Syt7 contains homologous mutations identified in our screen (R334C and E348K) in its native C2B sequence that indicates it is unlikely to engage the SNARE complex in the same manner as Syt1. Analysis of differences in mammalian Syt1 and Syt7 also argue for distinct C2B-dependent properties for the two proteins (*Xue et al., 2010*). It will be interesting in future studies to determine if differential SNARE binding between Syt1 and Syt7 contributes to their distinct roles in regulating synchronous and asynchronous phases of synaptic vesicle fusion.

Beyond the data defining an evolutionarily conserved role for Syt1-SNARE interactions in synaptic vesicle fusion, our work also provides insights into how dominant-negative C2B $Ca^{2+}$ binding mutations disrupt neurotransmitter release. Earlier studies in *Drosophila* indicated that mutating the $Ca^{2+}$ binding aspartate residues that line the C2B lipid penetration loops resulted in a dominant-negative disruption of release (*Lee et al., 2013*; *Mackler et al., 2002*; *Yoshihara et al., 2010*). Expression of this DN-Syt1 in *Drosophila* results in pupal lethality, and formed the basis for our initial screen for intragenic suppressors. It is important to note that the intragenic suppressor screen is unlikely to have hit all the key residues required for normal Syt1 function, as we specifically screened for mutations that block the DN effects of the mutant C2B protein as opposed to native Syt1 activity. The importance of defining how Syt1 with defective C2B $Ca^{2+}$ binding disrupts release is highlighted by recent findings that indicate this effect is conserved in humans. Two multigenerational families were identified with autosomal dominant mutations in the $Ca^{2+}$ binding pocket of the C2B domain of human Syt2, a Syt1 homolog enriched in the PNS (*Herrmann et al., 2014*; *Whittaker et al., 2015*). These patients display peripheral neuropathy and dysfunctional synaptic transmission at neuromuscular junctions (NMJs). From our genetic suppressor screen, it is clear that the mutant DN Syt1 protein is likely to engage and disrupt core mechanisms of Syt1 that lead to the reduction in neurotransmitter release. The screen revealed that altering SNARE binding (discussed above), C2A domain $Ca^{2+}$ binding (D229N), C2B domain lipid interactions (P363S), multimerization and/or C2A-C2B domain interactions (R250H), and the polybasic C2B surface regulated $Ca^{2+}$-independent lipid binding (K379E) all lead to a suppression of the dominant-negative effects on release. Each of these interactions also function during the normal synaptic vesicle cycle based on our in vivo rescue experiments with the point mutants expressed in the wildtype Syt1 protein lacking the $Ca^{2+}$ binding C2B mutations.

Given the positioning of the R250H mutation near the dimer surface observed in the Syt1 crystal structure, we explored this residue in more detail. Molecular modeling indicated the Syt1 dimer is likely to be stable in vivo (*Figure 5*), with dimer stability disrupted by the R250H amino acid substitution. If R250H disrupts the ability of mutant Syt1 to bind to and inactivate endogenous Syt1, that could prevent the ability of the mutant protein to interfere with release. Our prior studies of *Drosophila* Syt1 have suggested the protein is likely to function as a multimer (*Lee et al., 2013*; *Littleton et al., 1994*), consistent with a dimerization defect. An alternative model is that R250 plays a key role in stabilizing the interaction between the C2A and C2B domains (*Figure 4*), such that loss of an intramolecular C2A-C2B domain interaction blocks the DN effects on release. Single molecule experiments indicate that Syt1 in solution represents an ensemble of C2 conformations (*Choi et al., 2010*), and a subsequent modeling study suggested these conformations have tightly interacting C2 domains (*Bykhovskaia, 2015*). Recent observations indicate that a stable C2A-C2B interface is also important for Syt1 activity both in vitro and in vivo (*Evans et al., 2016*). Our molecular modeling indicates the normal C2A-C2B monomer stability of Syt1 is disrupted by R250H, consistent with a potential non-dimerization role for this residue as well. Consequentially, this mutation is likely to destabilize Syt1 conformations that are functional in vivo. However we cannot rule out other possible effects of R250 on Syt1 function. The R250 residue is situated in proximity to the C2A $Ca^{2+}$ binding

pocket and could potentially compromise C2A lipid binding as well. Future studies will be required to differentiate which of these functions of R250 are key for Syt1 activity. In vivo, Syt1 R250 fails to rescue the amount of synchronous fusion, although the timing of release and the suppression of asynchronous events are normal. There is also a striking defect in the ability of the R250 mutation to suppress the enhanced spontaneous release and to properly modulate synaptic vesicle density. As such, R250-mediated interactions regulate both the exocytotic and endocytotic functions of Syt1 in vivo. In summary, our genetic analysis of suppressors of Syt1 C2B $Ca^{2+}$ binding mutations have highlighted a key role for multiple Syt1 functions during neurotransmitter release, including an essential role for the Syt1-SNARE binding surface during the final stages of fusion.

## Materials and methods

### Genetic screen and dDrosophila stocks

*Drosophila melanogaster* were cultured on standard medium at 22°C. DNA constructs for *UAS-synaptotagmin 1 (syt1)*$^{C2B-D1,2N}$ encoding *Syt1*$^{D356N, D362N}$ (Syt1C2B$^{D1,2N}$) and *(syt1)*$^{C2B-D3,4N}$ encoding *Syt1*$^{D416N, D418N}$ (Syt1C2B$^{D3,4N}$) were obtained from N.E. Reist (Colorado State University, Fort Collins, CO). Transgenic strains were generated using standard microinjection into *white (w$^{-/-}$)* embryos performed by Duke University Model System Genomics (Durham, NC). *UAS-syt1* transgenes were expressed using a *GAL4* driver under the control of the pan-neuronal *elav* promoter. Twenty rounds of EMS mutagenesis (standard feeding with 25 mM EMS) with 1000 isogenized progeny of Syt1C2B$^{D1,2N \ or \ D3,4N}$ transgenic males was performed. Mutagenized males were crossed to isogenized elav$^{C155}$-GAL4 virgins and cultured on standard medium at 25°C. Males or female progeny surviving to the F1 generation were crossed to *white (w$^{-/-}$)*. After mating, genomic DNA was collected and sequenced via PCR with the following primer sets: CAACTGCAACTACTGAAATCTGCC (pUAST 5') and GTCACACCACAGAAGTAAGGTTCC (pUAST 3'). DNA for rescue with individual point mutants was generated using the QuikChange multisite-directed mutagenesis kit (Stratagene, Santa Clare, CA, USA) with the following primer sets:

| Mutant Oligo | Oligo Sequence |
| --- | --- |
| R250H-5'oligo | GACCAAGGTGCACCaCAAGACACTGAGTCCG |
| R250H-3'oligo | CGGACTCAGTGTCTTGtGGTGCACCTTGGTC |
| S332L-5'oligo | GGAGATATCTGCTTCTtGCTGCGCTACGTGCCG |
| S332L-3'oligo | CGGCACGTAGCGCAGCaAGAAGCAGATATCTCC |
| R334H-5'oligo | GATATCTGCTTCTCGCTGCaCTACGTGCCGACCGCCG |
| R334H-3'oligo | CGGCGGTCGGCACGTAGtGCAGCGAGAAGCAGATATC |
| K379E-5'oligo | GGCAAACGTTTGAAAAAGgAGAAGACAAGTGTCAAAAAATG |
| K379E-3'oligo | CATTTTTTGACACTTGTCTTCTcCTTTTTCAAACGTTTGCC |
| P363S-5'oligo | CGTGGGCGGACTGTCTGATtCATATGTGAAAGTTGCAATC |
| P363S-3'oligo | GATTGCAACTTTCACATATGaATCAGACAGTCCGCCCACG |

Wild type and mutant cDNAs were subcloned into a modified pValum construct with an N-terminal myc tag to allow tracking of protein localization in overexpressed animals containing endogenous Syt1 with the following primer sets:

GAATTCATGCCGCCAAATGCAAAATC (Syt1 EcoR1 5') and

TCTAGATTACAGATCTTCTTCAGAAATAAGTTTTTGTTCCTTCATGT

TCTTCAGGATCT (Syt1 3' c-myc Xba1). These constructs were injected into a yv; attP third chromosome docking strain by BestGene Inc. (Chino Hills, CA). All constructs allowed use of the Gal4/UAS expression system to express the transgenic proteins. *UAS-Syt1* transgenes were expressed using a *GAL4* driver under the control of the pan-neuronal C155 *elav* promoter in either control *white* or

*syt1* null (*syt1*$^{-/-}$) backgrounds. Null mutants lacking endogenous Syt1 were generated by crossing *syt1*$^{N13}$, an intragenic *Syt1* deficiency (*Littleton et al., 1994*), with *Syt1*$^{AD4}$, which truncates Syt1 before the transmembrane domain (*DiAntonio and Schwarz, 1994*).

## Electrophysiology

Postsynaptic currents from third instar male larvae at the wandering stage from the indicated genotypes were recorded at muscle fiber 6 of segment A3 using two-electrode voltage clamp with a −80 mV holding potential in HL3.1 saline solution as previously described (*Jorquera et al., 2012*). Final $Ca^{2+}$ concentration was adjusted to the desired level indicated in the text. For evoked and mini analysis, n refers to the number of NMJs analyzed, with no more than two NMJs analyzed per animal, and with animals derived from at least three independent experiments. For determining $Ca^{2+}$ cooperativity of release, 9 recordings were performed for each indicated concentration, with n indicating the total number of recordings in the 10 different $Ca^{2+}$ concentrations. Data acquisition and analysis was performed using Axoscope 9.0 and Clampfit 9.0 software (Molecular Devices, Sunnyvale, CA, USA). Motor nerves innervating the musculature were severed and placed into a suction electrode so action potential stimulation could be applied at the indicated frequencies using a programmable stimulator (Master8, AMPI; Jersalem, Israel).

## Electron microscopy and FM 1–43 loading

Electron microscopy and quantification of SV distribution were performed as previously described (*Akbergenova and Bykhovskaia, 2009*), with the exception that sections were cut at 60 nm thickness and stained with 2% uranyl acetate. For EM quantification, n refers to the number of NMJ boutons analyzed from at least three independent experimental animals. To assay endocytosis, preparations were stimulated for 0.5 or 1 min at 10 Hz in the presence of the lipophilic dye FM 1–43 (2,5 µm). Immediately after stimulation, preparations were rinsed in FM 1–43 free solution and incubated for 1 min with ADVASEP−7 (100 µm) to help remove non-internalized FM 1–43 dye. Preparations were fixed for 10 min in 4% paraformaldehyde, washed for 5 min and stained for 2 hr with anti-HRP Alexa647 antibody. Preparations were imaged using 63X water immersion lens after 10 min post staining wash. To estimate the level of FM 1–43 dye internalization, the HRP positive NMJ area was selected using Volocity 6.3 (Perkin Elmer) software and total FM 1–43 fluorescence was normalized by HRP area.

## Data analysis and statistics

Electrophysiology analysis was performed using Clampfit 10 software (Axon Instruments, Foster City). Statistical analysis and graphs were performed using Origin Software (OriginLab Corporation, Northampton, MA, USA). Statistical significance in two-way comparisons was determined by a Student's *t*-test, while one-way ANOVA nonparametric analysis was used when comparing more than two datasets. The P values associated with one-way ANOVA tests were adjusted P values obtained from a post hoc Tukey's multiple comparisons test or a Sidak's multiple comparison test as indicated. Appropriate sample size was determined using GraphPad Statmate. In all figures, the data is presented as mean ±SEM. Statistical comparisons are with control unless noted. An Excel file with all source data and statistical comparisons has been included as 'Source Data and Statistical Analysis' that contains individual spreadsheets labeled with the figure number and includes all primary data.

## Molecular modeling

Molecular modeling was performed as previously described (*Bykhovskaia, 2015*). The 2R83 Syt1 structure (*Fuson et al., 2007*) was used for an initial approximation of the Syt1 monomer and dimer. The optimized structure was placed in a water box (100 × 100×100 A for the monomer and 120 × 120×120 for the dimer) constructed using Visual Molecular Dynamics Software (VMD, Theoretical and Computational Biophysics Group, NIH Center for Macromolecular Modeling and Bioinformatics, at the Beckman Institute, University of Illinois at Urbana-Champaign). Potassium and chloride ions were added to neutralize the system and yield 150 mM KCl. Single point mutations were created using VMD. MD production runs were performed on the Anton supercomputer with Desmond software (*Shaw et al., 2009*) through the MMBioS (National Center for Multiscale Modeling of Biological Systems, Pittsburg Supercomputing Center and D.E. Shaw Research Institute). The initial

equilibration (100–200 ns) was performed employing NAMD Scalable Molecular Dynamics (*Phillips et al., 2005*) (Theoretical and Computational Biophysics Group, NIH Center for Macromolecular Modeling and Bioinformatics, at the Beckman Institute, University of Illinois at Urbana-Champaign) at the XSEDE (Extreme Science and Engineering Discovery Environment) Stampede cluster (TACC). Both NAMD and Desmond simulations were performed employing the CHARMM22 force field (*Mackerell, 2004*) with periodic boundary conditions and Ewald electrostatics in NPT ensemble at 300K with a time-step of 2 fs. The trajectory analysis was performed employing VMD and Vega ZZ (Drug Design Laboratory) software; PyMOL software was used for visualization, molecular graphics, and illustrations. In production runs, the time-step between trajectory points was 240 ps.

## Acknowledgements

This work was supported by NIH grants NS40296 and MH104536 to JTL, NIH grant MH099557 to MB and JTL, and AR063634 to RBS. We thank the Bloomington *Drosophila* Stock Center (NIH P40OD018537), the Developmental Studies Hybridoma Bank, Noreen Reist (Colorado State University) for providing cDNA, Yao Zhang for technical support and Steven Worthington (Institute for Quantitative Science, Harvard University) for statistical support. MD simulations were performed at the Anton supercomputer (DE Show Research and Pittsburg Supercomputer Center) and at XSEDE resources (Stampede supercomputer at TACC).

## Additional information

### Funding

| Funder | Grant reference number | Author |
| --- | --- | --- |
| National Institutes of Health | NS40296 | J Troy Littleton |
| National Institutes of Health | MH099557 | Maria Bykhovskaia |
| National Institutes of Health | AR063634 | Roger Bryan Sutton |
| National Institutes of Health | MH104536 | J Troy Littleton |

The funders had no role in study design, data collection and interpretation, or the decision to submit the work for publication.

### Author contributions

Zhuo Guan, Conceptualization, Data curation, Formal analysis, Investigation, Writing—original draft, Writing—review and editing; Maria Bykhovskaia, Data curation, Formal analysis, Funding acquisition, Investigation, Writing—original draft, Writing—review and editing; Ramon A Jorquera, Supervision, Writing—review and editing; Roger Bryan Sutton, Data curation, Formal analysis; Yulia Akbergenova, Data curation, Formal analysis, Investigation, Writing—review and editing; J Troy Littleton, Conceptualization, Formal analysis, Supervision, Funding acquisition, Writing—original draft, Project administration

### Author ORCIDs

Ramon A Jorquera, http://orcid.org/0000-0002-5460-1755
J Troy Littleton, http://orcid.org/0000-0001-5576-2887

### Decision letter and Author response

Decision letter https://doi.org/10.7554/eLife.28409.041
Author response https://doi.org/10.7554/eLife.28409.042

## Additional files

### Supplementary files

• Source data 1. Sample size (n), mean, SEM, and One-way Anova (and nonparametric) Sidak's multiple comparisons test are presented for the data in *Figure 9—figure supplement 1*.

DOI: https://doi.org/10.7554/eLife.28409.039

• Transparent reporting form
DOI: https://doi.org/10.7554/eLife.28409.040

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
