## [Decision Letter]

Thank you for submitting your article "A Synaptotagmin suppressor screen indicates SNARE binding controls the timing and Ca^2+^ cooperativity of vesicle fusion" for consideration by *eLife*. Your article has been favorably evaluated by a Senior Editor and three reviewers, one of whom is a member of our Board of Reviewing Editors. The reviewers have opted to remain anonymous.

The reviewers have discussed the reviews with one another and the Reviewing Editor has drafted this decision to help you prepare a revised submission.

Summary:

In this work, the authors identified residues of Syt1 that can suppress the dominant negative phenotype of certain calcium binding region mutants of the C2B domain. An intragenic screen was performed to find suppressors of the dominant negative C2B mutant. The authors then focused on those that did not involve a premature stop-codon or a deletion mutation, and that did not alter expression levels as assessed by Western analysis. Several groups of single-site mutants were analyzed in more detail. Moreover, for a selected subset, the mutants were expressed on a Syt1 null background. Several sensitive residues were found that localize to the polybasic region, a putative site of an intra-C2 domain contact, and the interface between Syt1 and the SNARE complex as found in the crystal structure by Zhou et al. (2015). The mutations that were tested in the Syt1-SNARE complex interface produced the strongest phenotype, including greatly decreased rapid synchronization and calcium cooperativity. This work provides new insights about the importance of residues of Syt1 that are not involved in calcium binding, but rather the interaction with the SNARE complex, the membrane, and putative intra-C2 domain interactions.

Essential revisions:

1) In the first paragraph of the subsection “Intragenic suppressors in Syt1 C2B^D1,2N^ rescue evoked release defects” the authors mention their observation that overexpression of Syt1 with mutations in both the C2A and C2B calcium binding sites did not cause dominant-negative defects in release, as also shown their previous work (Lee et al., 2013). However, Wu et al. recently found that simultaneous mutation of both C2A and C2B calcium binding sites are still dominant negative in mouse neurons, more so than the C2B mutant itself (D. Wu et al., Postsynaptic synaptotagmins mediate AMPA receptor exocytosis during LTP. Nature, 1-21 (2017). The authors are encouraged to discuss and speculate about the reasons for this interesting difference.

2) In addition to S332 (S279 in rat) and R334 (R281 in rat), residue Y391 (Y338 in rat), also plays a key role in the Syt1-SNARE interface found by Zhou et al. (2015). Indeed, according to Table 1 is also a suppressor of the dominant negative C2B mutant. Did the authors perform any additional experiments with the Y391 mutant?

3) The authors speculate that the R250H (R199 in rat) suppressor may be related to intra-C2 interactions. However, the crystal structure of Syt1 by Fuson et al. (2007) is only one possible conformation between Syt1 C2 domains. Single molecule FRET experiments by Choi et al. (NSMB 17, 318, 2010) showed a rather large distribution of intra-C2 conformations, of which the conformation observed by Fuson et al. (2007) is a major conformation, but not the only one. Thus, the relatively mild suppressor action by R250H may be related to existence of multiple conformations between the C2 domains. Another possible explanation may be related to the proximity of R250 to the calcium binding loops of the C2A domain: the mutant may affect the membrane binding affinity of the C2A domain, which may in turn influence the phenotype of the C2B dominant negative mutant. The authors are encouraged to discuss these possibilities, and perhaps tone down or even remove the rather speculative computer simulations.

4) Figure 3, Figure 6 and Figure 8: please provide recordings from UAS-R250SYT OE flies and some of the other isolated mutants in a wild type background. Does expression of these mutants in syt1 null background rescue lethality?

5) Figure 9. The authors claim that R250H does not rescue the enlarged vesicle size phenotype, can the authors provide a quantification of large vacuoles with a diameter greater than 100 nm in syt1 null animals? Does this mutation actually cause an endocytosis defect? They should provide additional evidence that the mutation R250H affects endocytosis (e.g. FM1-43).

6) As mentioned by the authors, Syt1 has an acknowledged role in SV endocytosis, and perhaps some of the mutations that decrease the total SV number impair proper endocytosis. Do the authors have any means of addressing this directly or indirectly for their mutations? And as a related concern, although gross trafficking to the synaptic bouton appears normal in these point mutants, can the possibility be excluded that there is less Syt1 on the SVs (perhaps due to inefficient retrieval from the plasma membrane), and this decrease drives some of the exocytic defects? And lastly, is it possible to quantify the synaptic abundance of Syt1 to see if there's a correlation with overall synaptic abundance and function?

7) Many interesting regulatory steps in the process of SV exocytosis can be parsed into either SV priming or SV fusion probability. Although none of the mutations analyzed by EM showed a detectable shift in the number of SVs docked immediately at the T-bar, there was no effort made to measure or comment on the size of the readily-releasable pool and the impact of these Syt1 mutations on its size. Is there a technical limitation that prevents using sucrose or depleting stimulus trains examine this?

8) In the calcium cooperative analysis of Figure 8, it would be useful to plot the fit coefficients in the blank space next to the figure for comparison. The major effect is R334H vs. WT and everything else is sort of in between. I'm not sure if those differences would be statistically different using a multiple comparisons test rather than T test, but the major interesting case was the R334H mutation anyway, and that effect looks highly significant. Also, what is the error range reported along with the fit value? Is that an uncertainty generating from the fitting algorithm or is that a standard error from fits to multiple replicates of the eEJC vs. Ca curves? And finally, why wasn't the P363S mutation examined for changes in cooperativity since this may directly impact the calcium-binding site?

9) In general, for the physiology and EM studies of the Syt1 mutant, the transgenic animals were not compared to wild-type animals, only the Syt1 null rescued with wild-type Syt1. The WT transgenic rescue is certainly the main appropriate control strain for comparisons but it would be nice to see an example of null rescue compared to a wild-type animal perhaps along with the initial rescue experiments in Figure 3.

---

## [Author Response]

Essential revisions:1) In the first paragraph of the subsection “Intragenic suppressors in Syt1 C2B^D1,2N^ rescue evoked release defects” the authors mention their observation that overexpression of Syt1 with mutations in both the C2A and C2B calcium binding sites did not cause dominant-negative defects in release, as also shown their previous work (Lee et al., 2013). However, Wu et al. recently found that simultaneous mutation of both C2A and C2B calcium binding sites are still dominant negative in mouse neurons, more so than the C2B mutant itself (D. Wu et al., Postsynaptic synaptotagmins mediate AMPA receptor exocytosis during LTP. Nature, 1-21 (2017). The authors are encouraged to discuss and speculate about the reasons for this interesting difference.

Although the basis of this species-specific difference is unclear, the C2A domain from *Drosophila* could be more essential for proper positioning of the DN C2B domain that would allow it to poison the fusion machinery, a property lost by simultaneous disruption of its Ca^2+^-activated lipid binding properties. There could, of course, be alternative explanations as well, but this would potentially explain the differences. We cite the Wu paper in our revised manuscript to note this difference, and provide a brief discussion.

2) In addition to S332 (S279 in rat) and R334 (R281 in rat), residue Y391 (Y338 in rat), also plays a key role in the Syt1-SNARE interface found by Zhou et al. (2015). Indeed, according to Table 1 is also a suppressor of the dominant negative C2B mutant. Did the authors perform any additional experiments with the Y391 mutant?

Indeed, Y391, as well as E348 and A455, came out of our screen as residues that appear at or near the recently identified Syt1-SNARE interface. We recovered four independent suppressors that alter Y391, so clearly this residue is critical for the DN Syt1 effect. However, we focused on S332 and R334 as representative of this group of Syt1-SNARE surface interaction residues, and did not do additional experiments on the other hits located on this surface of the protein. We list these hits in our table, and they support our conclusions on the importance of the Syt1-SNARE interface, but given the time involved in a detailed characterization of each mutant allele and the requirement to make a host of transgenic animals for each hit we characterize, we focused on picking representatives of each category to further analyze, in this case S332 and R334.

3) The authors speculate that the R250H (R199 in rat) suppressor may be related to intra-C2 interactions. However, the crystal structure of Syt1 by Fuson et al. (2007) is only one possible conformation between Syt1 C2 domains. Single molecule FRET experiments by Choi et al. (NSMB 17, 318, 2010) showed a rather large distribution of intra-C2 conformations, of which the conformation observed by Fuson et al. (2007) is a major conformation, but not the only one. Thus, the relatively mild suppressor action by R250H may be related to existence of multiple conformations between the C2 domains. Another possible explanation may be related to the proximity of R250 to the calcium binding loops of the C2A domain: the mutant may affect the membrane binding affinity of the C2A domain, which may in turn influence the phenotype of the C2B dominant negative mutant. The authors are encouraged to discuss these possibilities, and perhaps tone down or even remove the rather speculative computer simulations.

As suggested, Syt1 in solution is likely to represent an ensemble of C2 conformations. Both FRET experiments and MD simulations (Bykhovskaia, Biophys. J. 2015, 108:2507-20) agree with this view. Indeed, the simulations presented in our study for naïve Syt1 (Figure 4) also agree with this model, since they show a conformational transition for C2 domains within a time period of only 3 microseconds. Importantly, we also show that the R250H mutation has a drastic effect on the conformational dynamics of C2 domains. Specifically, as shown in Figure 4, within the same 3 microsecond time period, the mutated protein undergoes multiple conformational transitions, including separations and major rotations of the domains. It would be surprising if such major alterations in C2 domain dynamics had no effect on Syt1 function. Even if the functional Syt1 conformation differs from the one obtained by crystallography in Fuson et al. (2007), this conformation would still likely be destabilized by the R250H mutation, since this mutation enhances conformational transitions of the Syt1 C2 domains. We discuss this in more detail in the revised manuscript. We agree, however, that the R250H mutation could also modify membrane binding of C2A, as suggested by the referee. We included this alternative possibility in the revised manuscript.

4) Figure 3, Figure 6 and Figure 8: please provide recordings from UAS-R250SYT OE flies and some of the other isolated mutants in a wild type background. Does expression of these mutants in syt1 null background rescue lethality?

Indeed, it is very interesting to know if disrupting other interactions mediated by Syt1 can lead to a DN phenotype, as the reviewer suggests. To test our current mutations, we completed a new set of experiments where we overexpressed the Syt1 mutant constructs in a wildtype background. We found that none of the mutations we characterized in the current study (R250H, K379E, P363S, S332L and R334H) caused any significant changes in either evoked or spontaneous release compared to either controls or overexpression of wildtype Syt1. As such, mutations in the C2B Ca^2+^ binding pocket appear to be unique so far in their ability to poison the fusion machinery at the *Drosophila* NMJ. We have included this new data in the revised manuscript, and as a new figure (Figure 2—figure supplement 2).

In terms of whether the mutant proteins can alter the lethality of the *syt1* null, we performed a new set of experiments to examine their rescue ability. We include this new data in Figure 3—figure supplement 2, panel D. The wildtype Syt1 transgene robust rescues the lethality, with R250H and K379E also generating very significant rescue. In contrast, S332L has a very little rescue, with R334H and P363S providing no rescue at all. These effects on viability match very well with their ability to rescue the null physiology defects, as R334H and P363S have essentially no rescue of the synaptic release defects.

5) Figure 9. The authors claim that R250H does not rescue the enlarged vesicle size phenotype, can the authors provide a quantification of large vacuoles with a diameter greater than 100 nm in syt1 null animals? Does this mutation actually cause an endocytosis defect? They should provide additional evidence that the mutation R250H affects endocytosis (e.g. FM1-43).

As requested by the reviewers, we have quantified the EM phenotype of the *syt1* null in relation to the large vacuolar phenotype we observe. Whereas the wildtype rescue has 0.74 ± 0.16 vacuoles per µm^[2]^, *syt1* null mutants have significantly more (1.64 ± 0.28, p<0.01 vacuoles per µm^[2]^). The R250H rescue shows an even more dramatic phenotype, with 2.9 ± 0.5 vacuoles per µm^[2]^. Why is the phenotype even worse in the R250H mutant? We hypothesize that since the null releases so few vesicles on the exocytotic side, this accumulation of excess membrane has less opportunity to develop. The R250H has far more evoked release than the null, which we propose leads to a more robust defect in the endocytic process. We have included this new quantification in the Results section.

In addition, we examined FM 1-43 loading at 30 seconds and 1 minute during a 10 Hz stimulation train in the two mutants that show less synaptic vesicles by EM – R250H and K379E. Following 1 min of loading, labeled synaptic vesicles are distributed uniformly across the bouton periphery for *syt1* null animals rescued with wildtype Syt1. In contrast, *syt1* null mutants show reduced loading, with small clusters of vesicles distinguishable. The R250H and K379E rescue lines have diminished loading at 1 min compared to controls. This defect is even more robust during shorter stimulations (10 Hz for 30 seconds), where R250H and K379E rescue lines have reduced loading and show more punctate labeling than controls, consistent with the reduction in synaptic vesicles observed by EM. We have included this data as a new figure (Figure 9—figure supplement 1).

6) As mentioned by the authors, Syt1 has an acknowledged role in SV endocytosis, and perhaps some of the mutations that decrease the total SV number impair proper endocytosis. Do the authors have any means of addressing this directly or indirectly for their mutations? And as a related concern, although gross trafficking to the synaptic bouton appears normal in these point mutants, can the possibility be excluded that there is less Syt1 on the SVs (perhaps due to inefficient retrieval from the plasma membrane), and this decrease drives some of the exocytic defects? And lastly, is it possible to quantify the synaptic abundance of Syt1 to see if there's a correlation with overall synaptic abundance and function?

To address this point in more detail, we added FM 1-43 uptake assays as described in point 5 above. We also have completed more physiology on the two mutants where EM points to an endocytotic defect – R250H and K379E. We used a 10 Hz stimulation paradigm that we previously employed in *complexin* mutants to examine synaptic vesicle pool sizes (Jorquera et al., J. Neuro. 32:18234-18245, 2012). Three major functional pools of synaptic vesicles have been described at many synapses – the immediate releasable pool (IRP), the ready releasable pool (RRP) and the reserve pool (RP), including the *Drosophila* NMJ (Rizzoli and Betz, 2005). Vesicles from the IRP contribute to phasic neuronal responses and are associated morphologically with docked vesicles (Elmqvist and Quastel, 1965; Schneggenburger et al., 1999; Delgado et al., 2000). Vesicles from the RRP drive tonic responses and are mobilized during stimulation to prevent depletion at active zones. Synaptic vesicle recycling is largely responsible for sustaining release after mobilization, with a minor contribution from the RP (Heuser and Reese, 1976; Delgado et al., 2000; Zucker and Regehr, 2002). By examining the steady-state response following depletion of the IRP and RRP, we can estimate the effective recycling pool size. Using this method, we find that indeed both the R250H and the K379E mutant rescues display defects in the synaptic vesicle pool size compared to wildtype rescues. We have included this data, together with the FM 1-43 staining, in a new figure (Figure 9—figure supplement 1).

In terms of whether the mutants could alter the levels of Syt1 on synaptic vesicles, we have no way of directly testing that. Our immunocytochemistry shows the mutant proteins are trafficked normally to the NMJ and are found in the typical halo pattern at individual synaptic boutons, similar to what is normally seen for synaptic vesicle proteins (Figure 3—figure supplement 1). Therefore, we have no reason to suspect such a defect.

7) Many interesting regulatory steps in the process of SV exocytosis can be parsed into either SV priming or SV fusion probability. Although none of the mutations analyzed by EM showed a detectable shift in the number of SVs docked immediately at the T-bar, there was no effort made to measure or comment on the size of the readily-releasable pool and the impact of these Syt1 mutations on its size. Is there a technical limitation that prevents using sucrose or depleting stimulus trains examine this?

Sucrose use in *Drosophila* is quite variable and not a great tool due to muscle contraction issues associated with that high a level of release at the NMJ, so we have avoided it in all our previous studies. We can do depleting stimulus trains, as the reviewer points out, and as we have done in point 6 above. However, these are labor-intensive experiments, so we have not completed this analysis for the full array of mutants – we focused on those that displayed a vesicle density defect by EM (R250H and K379E) in the current study. We do plan a follow-up study on an additional subset of the lines, particularly those that disrupt the Syt1-SNARE interface, where we will perform the full analysis of pool sizes. We are currently combining these rescue lines with *complexin* mutant strains to determine how the Syt1-SNARE binding dynamics interfaces with Complexin-SNARE binding in terms of vesicle docking and priming, vesicle clamping and Ca^2+^ activation of the release machinery. It will be exciting to examine the interplay between these two key regulators of SNARE biology (Syt1, Cpx), and we plan on incorporating more exhaustive experiments on the physiology in this context.

8) In the calcium cooperative analysis of Figure 8, it would be useful to plot the fit coefficients in the blank space next to the figure for comparison. The major effect is R334H vs. WT and everything else is sort of in between. I'm not sure if those differences would be statistically different using a multiple comparisons test rather than T test, but the major interesting case was the R334H mutation anyway, and that effect looks highly significant. Also, what is the error range reported along with the fit value? Is that an uncertainty generating from the fitting algorithm or is that a standard error from fits to multiple replicates of the eEJC vs. Ca curves? And finally, why wasn't the P363S mutation examined for changes in cooperativity since this may directly impact the calcium-binding site?

This information has been added to source data files linked to each figure, these provide the raw data and the statistical analysis for every single experiment in our study, in keeping with *eLife*’s new emphasis on data reproduction. Every experimental result for each figure panel is provided in the relevant source data file. As requested by the reviewer, we have included all our source data for the cooperativity measurements, including Hill fits and errors. We did not include the P363S mutation in this analysis, as we think the effects of this mutation, based on its location in the C2B pocket, is likely to be similar to Ca^2+^ binding mutants we previously generated and characterized in this region (Yoshihara et al., 2010). For the remaining mutants, as the reviewer notes, we see some effect on cooperativity in all the lines, perhaps not surprising given how central Syt1 is for Ca^2+^ activation of fusion. However, the R334H mutation stands out in this regard, as it completely abolishes cooperativity, similar to the null mutant, and argues for a key role for the Syt1-SNARE interaction in driving an aspect of the cooperativity puzzle.

9) In general, for the physiology and EM studies of the Syt1 mutant, the transgenic animals were not compared to wild-type animals, only the Syt1 null rescued with wild-type Syt1. The WT transgenic rescue is certainly the main appropriate control strain for comparisons but it would be nice to see an example of null rescue compared to a wild-type animal perhaps along with the initial rescue experiments in Figure 3.

Indeed, we previously found that release is quite sensitive to Syt1 levels. To avoid genomic position effects on transgenic expression in our current study, we used site-specific transformation via the ΦC31 integrase system to place the mutant Syt1 rescues in the same genomic locus. Although this ensures equal transcriptional levels of our control and mutant rescues, the transgenes are expressed at lower levels compared to wildtype Syt1, which is why all of our controls use the Syt1 wildtype rescue for comparison. We have included a new western analysis of Syt1 expression levels in the rescue condition in Figure 3—figure supplement 1, panel E, that shows the transgenic proteins are expressed at similar levels to the wildtype rescue. However, they are only expressed at ~20% of the levels of endogenous Syt1 (see new Figure 3—figure supplement 2). We document the physiological effects of this reduction in the overall level of wildtype Syt1 in a new figure – Figure 3—figure supplement 2, where the amount of synaptic vesicle release is decreased, consistent with Syt1 levels being an important modulator of fusion. Although the overall amount of vesicle fusion is reduced compared to the levels of native Syt1, release is still fully synchronous in the wildtype rescue, and mini frequency elevation is full restored, as shown in Figure 3. The wildtype rescue also fully rescues viability of the null mutant as shown in Figure 3—figure supplement 3. For all our figures, we subsequently compared the mutant rescues with the wildtype rescue expressed from the same genomic insertion site.